# Deep uncertainties in shoreline change projections: an extra-probabilistic approach applied to sandy beaches

Rémi Thiéblemont[1], Gonéri Le Cozannet[1], Jérémy Rohmer[1], Alexandra Toimil[2], Moisés Álvarez-Cuesta[2], Iñigo J. Losada[2]

[1]Bureau de Recherches Géologiques et Minières "BRGM", French Geological Survey, 3 Avenue, Claude Guillemin, CEDEX, 45060 Orléans, France
[2]IHCantabria-Instituto de Hidráulica Ambiental de la Universidad de Cantabria, Parque Científico y Tecnológico de Cantabria, Calle Isabel Torres 15, 39011 Santander, Cantabria, Spain

*Correspondence to*: Rémi Thiéblemont (r.thieblemont@brgm.fr)

**Abstract.** Global mean sea-level rise and its acceleration are projected to aggravate coastal erosion over the 21[st] century, which constitutes a major challenge for coastal adaptation. Projections of shoreline retreat are highly uncertain, however, namely due to deeply uncertain mean sea-level projections and the absence of consensus on a coastal impact model. An improved understanding and a better quantification of these sources of deep uncertainty are hence required to improve coastal risk management and inform adaptation decisions. In this work we present and apply a new extra-probabilistic framework to develop shoreline change projections of sandy coasts that allows considering intrinsic (or aleatory) and knowledge-based (or epistemic) uncertainties exhaustively and transparently. This framework builds upon an empirical shoreline change model to which we ascribe possibility functions to represent deeply uncertain variables. The model is applied to two local sites in Aquitaine (France) and Castellón (Spain). First, we validate the framework against historical shoreline observations and then develop shoreline change projections that account for possible (although unlikely) low-end and high-end mean sea-level scenarios. Our high-end projections show for instance that shoreline retreats of up to 200m in Aquitaine and 130m in Castellón are plausible by 2100, while low-end projections revealed that 58m and 37m modest shoreline retreats, respectively, are also plausible. Such extended intervals of possible future shoreline changes reflect an ambiguity in the probabilistic description of shoreline change projections, which could be substantially reduced by better constraining SLR projections and improving coastal impact models. We found for instance that if mean sea-level by 2100 does not exceed 1m, the ambiguity can be reduced by more than 50 %. This could be achieved through an ambitious climate mitigation policy and improved knowledge on ice-sheets.

## 1 Introduction

Global mean sea level has risen over the period 2006-2015 at a rate more than twice larger than over the whole 20[th] century, and is projected to continue rising for the centuries to come (Oppenheimer et al., 2019). This inevitable sea-level rise (SLR) will exacerbate risks in coastal areas, notably erosion and flooding. Recent analysis of satellite derived shoreline changes have revealed that a quarter of world's sandy beaches are eroding (Luijendijk et al., 2018) and that the overall surface of eroded land recorded over the period 1984-2015 (about 28,000 km²) is twice larger than the surface of gained land (Mentaschi et al., 2018). This situation is projected to worsen with climate change (Ranasinghe, 2016;Vousdoukas et al., 2020). Yet, future coastal retreat projections are highly uncertain, reflecting the deep uncertainties of future sea-level rise projections and of coastal impact models (Le Cozannet et al., 2019a;Athanasiou et al., 2020;Ranasinghe, 2020;Cooper et al., 2020;Vershuur et al., 2020). An

improved understanding and a better quantification of these sources of uncertainty are required to improve coastal risk management and inform adaptation decisions (Stephens et al., 2017).

Since the release of the Fifth Assessment Report (AR5) of the Intergovernmental Panel on Climate Change (IPCC) (Church et al., 2013), SLR projections by 2100 have been reassessed upwards and the range of uncertainty has enlarged for high greenhouse gas emissions scenarios (Oppenheimer et al., 2019). This update of IPCC SLR projections is due to the consideration of Marine Ice Sheet Instabilities (Joughin et al., 2014;Rignot et al., 2014). Hence, the IPCC Special Report on The Ocean and Cryosphere in a changing Climate (SROCC) revised the median

SLR by 2100 to 0.84 m for the RCP8.5 scenario (instead of 0.74 m of the AR5) and the upper limit of the likely range jumped to 1.1 m (instead of 0.98 m).

Besides, the SLR projections delivered by the IPCC do not cover the whole range of uncertainties. In fact, future ice-sheet contributions remain deeply uncertain, as a collapse of the west Antarctic ice-sheet during the 20$^{th}$ century cannot be excluded yet (DeConto and Pollard, 2016;Edwards et al., 2019). Hence, a possibility for future SLR

projections to lie above or below the IPCC likely range remains. Evidence for the possibility of large ice-sheets contribution to sea-level rise include e.g. physical modelling of melting processes (DeConto and Pollard, 2016) and structured expert judgment (Bamber et al., 2019). For example, Bamber et al. (2019) found that SLR could exceed 2m by 2100 for a high emission scenario (lying within the 90% uncertainty bounds), reflecting at least the absence of consensus within the community of glaciologists. Importantly, the gravitational effects of large ice-

sheets mass losses mean that sea-level rise would exceed the global mean along most inhabited shorelines. For example, Thiéblemont et al. (2019) showed that given the current ocean and cryosphere physical-based projections, the SLR could possibly – although unlikely - be as high as 1.9 m off the coasts of Western Europe by 2100 under the RCP8.5 scenario. The deep uncertainty associated with future regional sea-level change reflects the incomplete understanding of the underlying physical processes but also the uncertain magnitude of the global

warming in the future.

Coastal impact models used to project the shoreline change response to sea-level rise are another major source of uncertainty (Ranasinghe, 2016, 2020;Toimil et al., 2020). Shoreline changes are controlled by multiple hydro-sedimentary processes that interact with each other and operate at multiple timescales (ranging from one day to several decades) and spatial scales (Stive et al., 2002). Processes driving shoreline change are also extremely

variable from one beach segment to another, making very challenging the development of a standardized process-based modelling framework. Although numerical models have demonstrated significant skilful predictions of shoreline changes (Montaño et al., 2020), their use is generally restricted to local applications where high resolution and high accuracy data (e.g. topo-bathymetry, nearshore hydrodynamics, sediment characteristics, etc.) are available (Robinet et al., 2018;Enríquez et al., 2019). At large scale (generally > 500 km), assessments of

shoreline change projections (Hinkel et al., 2019;Thiéblemont et al., 2019;Vousdoukas et al., 2020;Athanasiou et al., 2020) rely widely on the Bruun rule, a two-dimensional cross-shore model that predicts landward retreat of the shoreline in response to SLR assuming a conserved equilibrium beach profile (Bruun, 1962). Nonetheless, the usefulness of the Bruun rule as a predictive tool is highly debated, notably with regard to its lack of validation against observations, robustness and general applicability, as beach segments generally do not meet the

assumptions for the Bruun rule application (Stive, 2004;Cooper and Pilkey, 2004;Ranasinghe and Stive, 2009;Ranasinghe, 2016;Cooper et al., 2020). For example, several studies found that the Bruun rule tends to

provide substantially higher shoreline retreat projections than physics-based probabilistic shoreline change models (Ranasinghe et al., 2012;Toimil et al., 2017;Le Cozannet et al., 2019a;Enríquez et al., 2019).

This deep uncertainty context inherent to shoreline change projections is a major challenge for coastal management and adaptation decision. Hinkel et al. (2019) showed that different kinds of information on sea-level projections is required depending namely on the time horizon of coastal decision adaptation and on the degree of uncertainty tolerance of users. For medium to high uncertainty tolerance, probabilistic projections are particularly well suited to identify the adaptation alternative that has the best-expected outcome (Nicholls et al., 2014;Budescu et al., 2014). In contrast, when uncertainty tolerance is low, robust decision-making is preferable, which implies testing adaptation options against any plausible scenarios; hence considering high-end (Hinkel et al., 2015;Kopp et al., 2017;Stammer et al., 2019;Hinkel et al., 2019) and low-end (Le Cozannet et al., 2019b) projections (or scenarios), which explore plausible — although unlikely — upper and lower tails sea-level scenarios beyond the likely range, respectively. Although the literature above has focused on sea-level rise information needs, the same type of information is needed for its coastal impacts (Rohmer et al., 2019), raising the need for a framework allowing to propagate and analyse deep uncertainties from sea-level rise to its impacts.

To develop shoreline change projections that meet the needs of users with various risk tolerance different future scenarios need to be developed and combined with a large variety of sources of uncertainty. Two types of uncertainty need to be considered (Beven et al., 2018;Toimil et al., 2020): intrinsic uncertainty (also called aleatory), which is inherent to the considered process (e.g. internal variability) and knowledge-based uncertainty (also called epistemic), which stems from information incompleteness or lack of knowledge (incl. deep uncertainties). To date, both types of uncertainties have been addressed mainly using the tools provided by the probability theory and occasionally used in combination with expert knowledge (especially for sea-level projections (Oppenheimer et al., 2019;Bamber et al., 2019)). Yet, several studies have pointed out that the use of probabilities merges the different uncertainty types in a single format and can in turn induce an appearance of overconfidence in uncertainty quantification (Le Cozannet et al., 2017;van der Pol and Hinkel, 2019;Bakker et al., 2017;Rohmer et al., 2019). Such misleading effect can have serious impact on coastal risk management and planning. To overcome this disadvantage of the classical probabilistic setting, alternative mathematical representation methods have been developed (see a comprehensive overview by Dubois and Guyonnet (2011)). These are termed extra-probabilistic because they avoid the selection of a single probability law by bounding all the possible probability models consistent with the available data. The added value of these approaches has been discussed for global SLR projections (Le Cozannet et al., 2017) or to assess local flood impact (Rohmer et al., 2019) but has never been used in the context of coastal erosion to our knowledge.

In this paper, we build on the extra-probabilistic framework of uncertainty to develop a new and versatile modelling framework to project future shoreline changes of sandy beaches. This framework enables coastal risk managers to account exhaustively and transparently for uncertainty of different kinds (aleatory and epistemic) and more specifically for deep uncertainty by providing the necessary tools to quantify it (via the definition of high-end and low-end scenarios) to support various decision contexts. Section 2 describes the shoreline change extra-probabilistic framework development. Section 3 describes the physical characteristics of the two study sites and the associated data. In section 4, we validate the shoreline change modelling framework against historical records and then use them for future projections. Our results are further discussed in section 5.

**2 Extra-probabilistic framework for shoreline change projections**

**2.1 Extra-probabilistic framework: general principle**

Uncertainty representation consists of modelling the available knowledge, i.e. selecting the most appropriate mathematical tools and procedures for representing the available data/information while "accounting for all data and pieces of information, but without introducing unwarranted assumptions" (Beer et al., 2013). When a large number of observations is available, a probability distribution can be inferred from data/observations. In our case, this applies for instance to the mean sea-level in the Bay of Biscay over the recent historical period, for which several observational records from tide gauges exist. In situations where the data and information are very scarce, imprecise, vague, even incomplete (i.e. an environment of imperfect knowledge (Beer et al., 2013)), selecting an appropriate probability law can be ambiguous. The later issue is referred to as deep uncertainties in the literature and can be addressed quantitatively by extra-probabilistic methods (Dubois and Guyonnet, 2011).

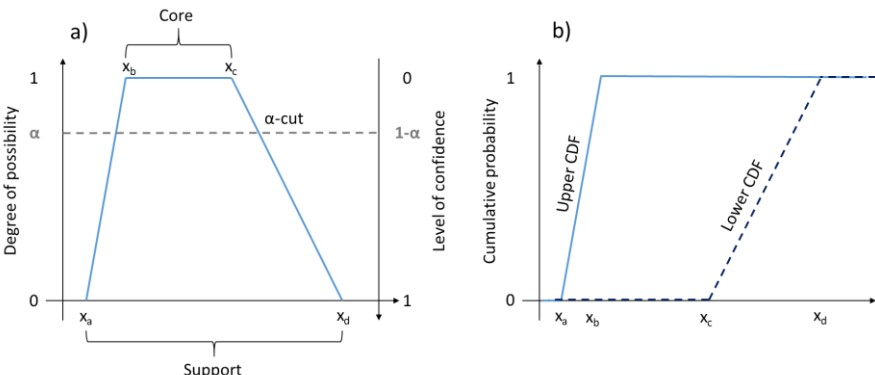

**Figure 1: Example of (a) trapezoidal possibility distribution and (b) its translation into a probability-box.**

Extra-probabilistic theories of uncertainty recognize that several probabilistic laws may exist given the piece of information available. Instead of providing a single uncertainty (probabilistic) model, they deliver sets of plausible probabilistic models. In the present study, we use the possibility theory to represent uncertainties of deeply uncertain variables (Dubois and Prade, 1988). The basic ingredient is the interval used for representing experts' knowledge. In most cases, however, experts may provide more information by expressing preferences within this interval. Such "nuanced" information can be conveyed using the possibility distributions, denoted $\pi$ (Dubois and Prade, 1988), which describes the more or less plausible values of some uncertain quantity. The intervals defined as $\pi_\alpha = \{e, \pi(e) \geq \alpha\}$ are called $\alpha$-cuts. They contain all the values that have a degree of possibility of at least $\alpha$ (lying between 0 and 1). The example of $\alpha$-cut on a trapezoid possibility distribution is shown on Fig. 1a. The interval for $\alpha=0$ and $\alpha=1$ is called the support and the core, respectively. The $\alpha$-cuts formally correspond to the confidence intervals 1-$\alpha$ as traditionally defined in the probability theory, i.e $\mathrm{Prob}(e \in \pi_\alpha) \geq 1 - \alpha$. Thus, a possibility distribution can be interpreted as a set of nested intervals, each of them being assigned with a level of confidence 1-$\alpha$. A possibility distribution then encodes a family of probability laws (Dubois and Prade, 1992), i.e. a probability-box limited by an upper probability bound called the possibility measure $\prod(e \in E) = \sup_{e \in E} \pi(e)$

(upper cumulative probability bound on Fig. 1b) and a lower probability bound called the necessity measure $N(e \in E) = \inf_{e \notin E}(1 - \pi(e))$ where E represents a specific interval (E=]1.0, +∞[ for instance). This link between

probabilistic and possibilistic theories was exploited by Le Cozannet et al. (2017) to derive a possibility
distribution to represent uncertainties on GSLR by 2100 conditional on RCP8.5 scenario.

**2.2 Setting-up shoreline change projections within the extra-probabilistic framework**

In principle, the extra-probabilistic framework can be used with any shoreline change model. In this study, we
adopt the perspective of coastal adaptation practitioners that generally rely on empirical models that extrapolate

observed shoreline changes to anticipate better their future evolution (Peter et al., 2003;Le Cozannet et al.,
2019a;Vousdoukas et al., 2020;Cowell et al., 2003). In the absence of estuaries or other major sediment sources
or sinks, our empirical model expresses shoreline change $\Delta S$ following Eq. (1):

$$\Delta S = S_t - S_{t0} = \frac{\Delta RSLC}{\tan \beta} + Lvar + n \cdot Tx \quad , \tag{1}$$

where $S_t - S_{t0}$ expresses the change in shoreline position in the cross-shore direction from reference time t$_0$ to

time t, $\frac{\Delta RSLC}{\tan \beta}$ quantifies the contribution of sea level rise to shoreline changes, which takes the form of the Bruun
rule (Bruun, 1962); $Tx$ is the linear trend of shoreline changes over multi-decadal timescales and *n* the number of
years relative to the baseline; $Lvar$ characterizes the interannual-to-decadal variability of shoreline change:
typically, $Lvar$ would quantify how the shoreline can depart from a mean position due to e.g., seasonal cycles or
the chronological sequence of storms and calm period. These terms, which are described further below, include

intrinsic and knowledge uncertainties that need to be adequately represented as input and then propagated. The
flowchart on Fig. 2 displays the three steps to develop shoreline change projections within the extra-probabilistic
framework.

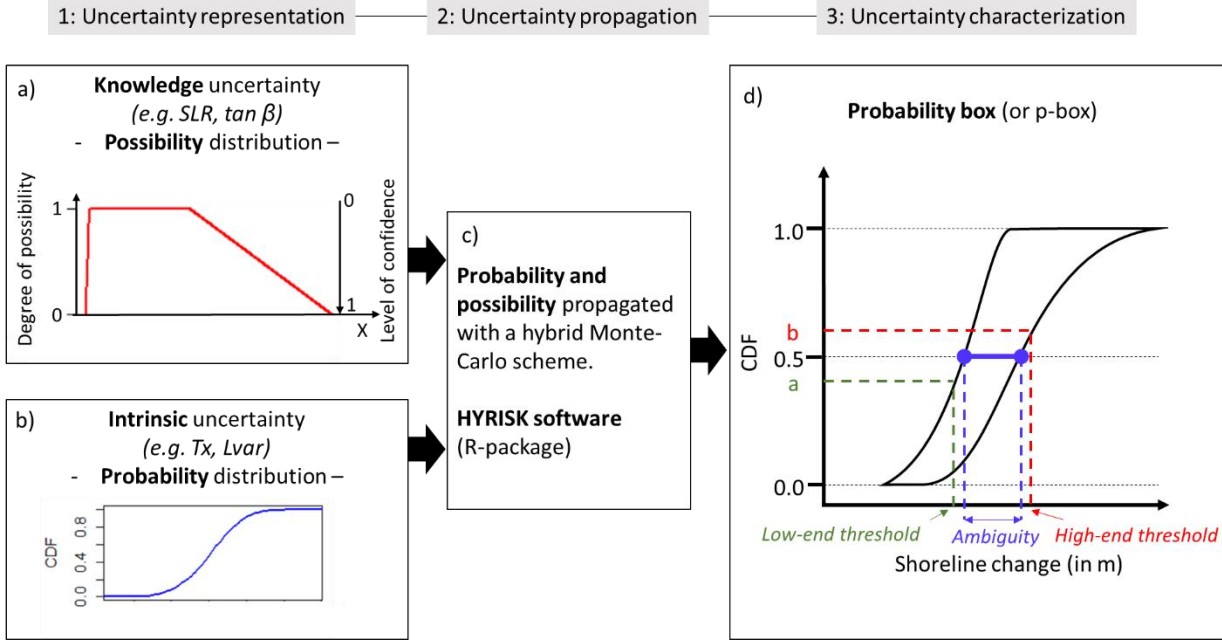

**Figure 2: Schematization of the framework used herein to perform future shoreline change projections within the extra-**
**probabilistic theory.**

As a first step (Fig. 2a,b), uncertainty distribution of inputs are constructed. For instance, in Eq. (1), $n \cdot Tx$ and
$Lvar$ are both derived from past shoreline change observations. Note that $Tx$ and $Lvar$ terms do not describe a
single physical process but rather a combination of processes that operate at different timescales including waves

climates, sediment budgets, effects of longshore gradients in sediment transport or anthropogenic actions. These processes are recognized complex and difficult to model with reduced complexity models (Montaño et al., 2020;Vitousek et al., 2017). By using the empirical model, our objective is to reproduce the observed trends and modes of variability without trying to model the physical processes explicitly, while keeping a low computation time (see Helgeson et al. (2020) for a broader discussion of this approach). The term $n \cdot Tx$ is the product of the number of years $n$ since the reference year and the multi-decadal linear trend $Tx$ derived from observations after substracting the effect of sea-level rise. In the case where multiple observations are available per year (typically when analysing shoreline changes retrieved from satellite imagery), the linear model used to derive $Tx$ is applied on annual means and weighted by the number of samples per year to account for the irregularity of the temporal sampling (see e.g. Fig. 4b). The residuals of the linear regression to compute $Tx$ are then used to derive $Lvar$. We sample residuals that are distant by a gap of N years (with $1 < N < 10$ as we focus on interannual-to-decadal timescales) and compute their standard deviation. This procedure is repeated for all possible combinations of residuals separated by N years. Finally $Lvar$ is determined as the maximum standard deviation value obtained among all samples. Note that $Lvar$ is found to maximize for $N \geq 5$ years. Since $Tx$ and $Lvar$ are derived assuming that errors of the linear regression are normally distributed, they are both prescribed as probability distributions.

In contrast, terms accounting for future sea level (ΔRSLR) and its impact on shoreline change (1/tan β) are both sources of deep uncertainty and are therefore too imprecise given the current knowledge to be constrained by probability distributions. For instance, to reflect the full range of current uncertainty, ΔRSLR should consider projections that are either below or beyond the likely-range provided by the IPCC, but for which probability are not well established. Regarding the coastal impact model, under the Bruun rule (Bruun, 1962), tan β corresponds to the slope of the active profile from the depth of closure to the top of the upper shoreface. The Bruun rule underlying assumptions include considering that sediment transport only occurs perpendicularly to the shoreline, thus neglecting any tri-dimensional variability, and assuming that the coastal profile is an equilibrium profile that has uniform sediment size. An alternative to the Bruun rule was proposed through the Probabilistic Coastline Recession (PCR) model (Ranasinghe et al., 2012). The PCR model quantifies sediment losses at the dune toe during storms, as well as the nourishment of the dune by aeolian sediment transport processes between storms. Given a certain amount of sea-level rise, the response of the PCR model is less erosive than the Bruun rule by one order of magnitude. While the use of the PCR model is rather expensive computationally, Le Cozannet et al. (2019a) demonstrated that, in a first approximation, the equilibrium response of the PCR model can be emulated in Eq. (1) by replacing the nearshore slopes (or Bruun slopes) by the foreshore slopes. Bruun and PCR models are however both difficult to validate because of the scarcity of coastal data and the complexity of the hydrosedimentary processes involved. This constitutes one of the source of deep uncertainty. Hence, to reflect the absence of consensus on coastal erosion induced by sea-level rise, neither surrogate PCR model nor Bruun rule should be discarded in our uncertainty propagation. To account for the limited knowledge of future sea level and its impact on shoreline change, we construct ΔRSLR and 1/tan β terms as trapezoidal possibility distribution (see also sections 4a and b).

As a second step (Fig. 2b), to propagate the heterogeneous uncertainty nature of the terms in Eq. (1), we used the HYRISK R package (Rohmer et al., 2017). HYRISK software is designed to jointly propagate probability and possibility by implementing the hybrid Monte-Carlo scheme, named Independent Random Sampling (IRS) algorithm developed by Baudrit et al. (2005). The IRS algorithm combines random sampling of the inverse of the

cumulative probability distribution functions for random parameters and the α-cuts (intervals associated to a level of confidence of 1-α) from the possibility distributions. More detail on the IRS algorithm is provided in the Appendix A. The result of the propagation procedure takes the form of random intervals that can be summarized by pair of upper and lower cumulative probability distributions (CDFs), which allows constructing probability boxes (or p-boxes, final step) based on the formal setting introduced by Baudrit et al. (2007).

Fig. 2d shows a typical example of shoreline change uncertainty propagation presented in the form of a p-box. The p-box is bounded to the left and right by the upper and lower CDF, respectively. The area enclosed within these two bounds includes all possible distributions of shoreline changes and characterize the full range of aleatoric and epistemic uncertainties. Epistemic uncertainty is represented by the breadth between the upper and lower CDF, whereas aleatory uncertainty is represented by the overall tilt of the p-box. The gap between the upper and lower CDF can be considered as "what is unknown" and represents the imperfect state of knowledge (Rohmer et al., 2019). To quantify this deep uncertainty, we use an indicator termed as "ambiguity" and defined as the width (in meter) between medians of the upper and lower CDF. In addition, we define the low-end threshold (i.e. minimum adaptation needs, shown in green) as the shoreline change value for which there is a chance smaller than $a$ to be reached under the less impacting (i.e. upper) CDF. In other words, the low-end value corresponds to a threshold, which is very likely to be exceeded. Finally, we define the high-end threshold (i.e. high risk-adverse applications, shown in red) as the value below which there is still more than $b$ chance for the projections to hold under the most impacting (i.e. lower) CDF. In this case, the high-end threshold corresponds to a value which can be possibly but unlikely exceeded. As an example, we define $a$ and $b$ as 0.4 and 0.6, respectively, although these thresholds are meant to be tailored to user needs depending on their risk aversion.

**3 Case studies and data**

In this work, the extra-probabilistic approach to perform shoreline change projections is applied in two coastal sites where the coastline is largely dominated by sandy beaches (Fig. 3) but (i) for which we have different sources of shoreline change observations and sampling, and (ii) that have highly distinct geomorphologic and hydrodynamical characteristics. Thereinafter, positive and negative values represent erosion and accretion, respectively, with respect to the baseline (i.e. 2015 for site 1 in Aquitaine and 2020 for site 2 in Castellón).

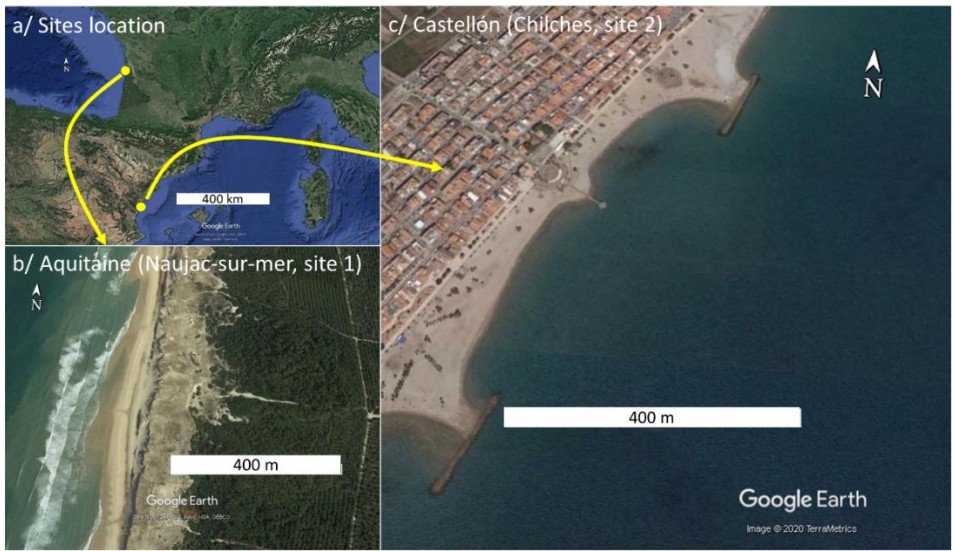

**Figure 3: Location of the two case studies on the Aquitaine (site 1, panel b) and Castellón (site 2, panel c) coasts. Basemaps are from Google Earth.**

**3.1 Case studies and shoreline change observational records**

The first site studied (Site 1) is located in the municipality of Naujac-sur-Mer, which belongs to the Aquitaine coast (Fig. 3). The Aquitaine is a 230 km long sandy coast located in south-western France constituted by high-energy meso-macrotidal open beaches, which are backed by coastal dunes with a typical height ranging between 15 and 20 m and a width larger than 100 m. The characteristic sediment of this coast is well-sorted sand, of medium to fine grain size between 250 and 300μm. Observational records of spatial and temporal shoreline change along the Aquitanian coast have been retrieved by Castelle et al, (2018). Their shoreline change dataset was generated based on 15 geo-referenced orthomosaics photos to examine long-term shoreline change from 1950 to 2014 along 270 km distributed over 2861 transects. They found a spatially averaged erosion trend of 1.1 m/year derived throughout the Aquitanian coast with maximum retreat (accretion) rates of 11 (-6) m/year. Here, the site studied (Fig. 3b) has been chosen to avoid influence of estuarine processes. Its observational records of shoreline change are shown on Fig. 4a. We found for this individual profile an erosion trend of 0.82 m/year, which is close to the time and Aquitanian spatially averaged erosion trend of 1.1 m/year.

The second site studied (Site 2) is in the Chilches municipality, located in the Mediterranean coast of Spain in the province of Castellón (Fig. 3c). The current coastal morphology in this area is highly conditioned by a succession of anthropic actions that started at the beginning of the 20th century. The construction of the ports of Castellón, Burriana and Sagunto completely blocked the northern contribution of sediments to downdrift of the structures. As a result, the coast shifted from the state of dynamic equilibrium with intense longshore transport and continuous sediment intake to imbalance, with the same longshore transport intensity but without any sediment contribution updrift. This resulted in the chronic recession of the beaches sheltered by the structures, and the accretion of the beaches located downdrift. Besides, the real estate boom that occurred in the second half of the 20th century exacerbated such imbalance, giving rise to constructions on beaches that were already in decline. Subsequently, to try to solve this problem, more actions were taken, including the construction of seawalls and jetties and replenishments. In their natural state, these are beaches of fine to medium sand with $D_{50}$ between 0.2-0.35 mm. Shoreline evolution in the Castellón-Sagunto stretch was retrieved using the CoastSat toolkit (Vos et al., 2019b) based on monthly or bimonthly observations from Landsat 5, Landsat 8 and Sentinel 2. CoastSat has been shown to have a particularly high accuracy in microtidal environments (Vos et al., 2019a). For the Castellón-Sagunto stretch, the dataset retrieved by CoastSat has been validated against discrete profile surveys at some specific sites. The shoreline evolution over the period 1989-2019 for the profile studied in Chilches is shown on Fig. 4b. Over the 31-year period, 859 shoreline positions (orange timeseries) were retrieved for this profile, with an average of 25 (70) observational records per year before (after) 2017. The profile shows an average coastline retreat of 0.6 m/year over the period 1989-2019.

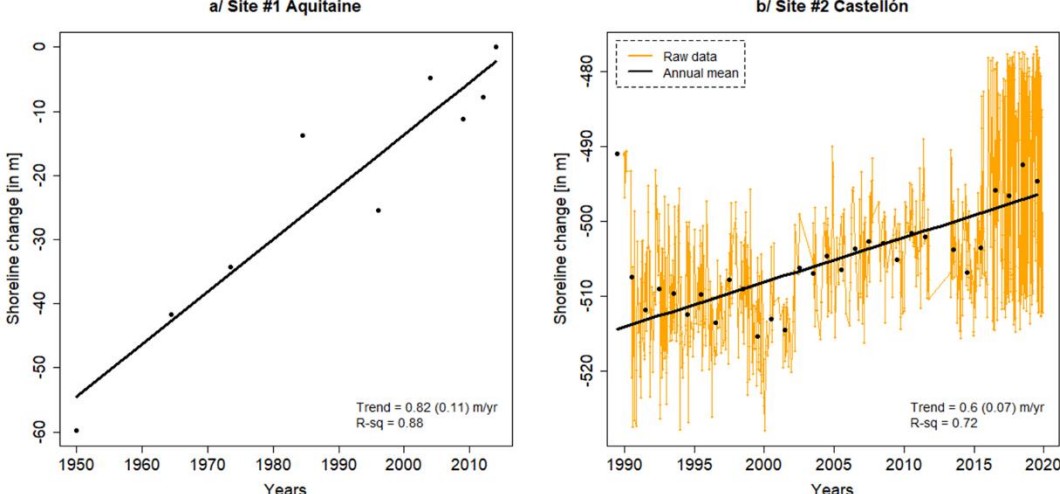

**270**

**Figure 4: Observed shoreline change evolution in (a) site #1 in Aquitaine and (b) site #2 in Castellón. In Castellón, 859 available observations (orange line) and annual averages (black dots) are shown. The black line shows the linear trends calculated from the annual averages and their standard error are written in brackets. Note that, by convention, positive trends value indicates shoreline retreat.**

**275** In Aquitaine (Site 1), topographic and bathymetric surveys recorded nearshore slopes comprised in the range 1.2%-1.5% that can occasionally be as mild as 1% (Bernon et al., 2016), and slopes of the upper shoreface that can be as steep as ~10% (Bulteau et al., 2014). In Castellón, beach slopes have been determined by combining two datasets: a topography dataset from the Spanish Geographic Institute (Instituto Geográfico Nacional, IGN); and a bathymetry dataset from the Spanish Ministry for the Ecological Transition and Demographic Challenge

**280** (Ministerio para la Transición Ecológica y el Reto Demográfico, MITERD). Specifically in Site 2, a nearshore slope of 3.1% was retrieved.

### 3.2 Historical sea level and projections

For both sites, the absolute sea-level time evolution in the past is constructed from tide gauge records which are corrected from vertical land motion based on Global Navigation Satellite System (GNSS) station records or global

**285** isostatic adjustment models (if GNSS stations are not found nearby). For the Aquitanian coastline, Bay of Biscay past sea-level is derived as the average of 15 stations available in the Permanent Service for Mean Sea Level (PSMSL) (see Le Cozannet et al. (2019a) for details). For the Castellón area, and in general along the east coast of Spain, tide gauges records provided in PSMSL are short and do not span the entire time period considered here (i.e. 1989-2019). Therefore, we relied on the Marseille tide gauge (GNSS corrected) records calculated over the

**290** period 1989-2019.

To obtain RSLC regional projections and their related uncertainty, we combine the future regional contributions of sterodynamic effects, melting of mountain glaciers and ice sheets, land water and glacial isostatic adjustment (Slangen et al., 2012;Slangen et al., 2014;Gregory et al., 2019) following the procedure described in Chapter 13 of the IPCC AR5 (Church et al., 2013). Regional projections of the sterodynamic component, which corresponds

**295** to changes in ocean density and circulation corrected from the inverse barometer effect, are derived from the outputs of the global climate model simulations performed within the 5[th] phase of the Coupled Model

Intercomparison Project (CMIP5). Note however that our sterodynamic projections slightly deviate from the IPCC AR5 and SROCC procedure:

- among the 21 CMIP5 models available, MIROC-ESM and MIROC-ESM-CHEM models are discarded as they project anomalously large sterodynamic component in the Atlantic and North Sea areas, such that if both models are retained, the distribution of the multi model ensemble is no longer Gaussian(Thiéblemont et al., 2019). Le Cozannet et al. (2019) have also shown that by 2100, the global-mean thermosteric sea-level rise of these two models (0.5 m for the RCP8.5 scenario) exceeds the median global-mean thermosteric sea-level rise of all other models (0.3 m) beyond 5 sigma;

- in the semi-enclosed basins (e.g. Mediterranean Sea), the rather coarse resolution of AOGCMs prevents an accurate representation of small-scale processes (e.g. water exchange at Gibraltar), which in turn affects regional sea-level estimates (Marcos and Tsimplis, 2008;Slangen et al., 2017). The Mediterranean sterodynamic sea-level projections are therefore estimated by relying on those of the Atlantic area near Gibraltar, which is the Mediterranean Sea entry point.

For other mass contributions to sea level (i.e. glaciers, ice sheets, land water), regional changes are obtained by downscaling global estimates and their uncertainty using barystatic-GRD fingerprints (Gregory et al., 2019). Finally, the regional RSLC likely range is computed as the square root of the sum of the squares of each regionalized component's likely-range (except for contributions that correlate with global-mean air temperature – see Church et al. (2013) for detail). Sterodynamic sea-level projections and barystatic-GRD fingerprints are available from the Integrated Climate Data Center of the University of Hamburg (http://icdc.cen.uni-hamburg.de/) (Carson et al., 2016).

To account for deep uncertainty in SLR, we do not only restrict to likely-range estimates but also consider low- and high-end estimates for the design of our RSLC projections. There is low confidence that sea-level rise can reach such values, yet, they cannot be discarded. There is no unique approach to design low- and high-end scenarios as reflected by the recent literature that abounds in sea-level projections that explicitly included high-end scenarios with various assumptions and methods (e.g. Wong et al., 2017; Le Bars et al., 2017; Stammer et al., 2019). Here, we choose to follow a consistent approach for low- and high-end scenarios, that is:

- low-end projections are based on the most conservative estimates of glaciers and ice-sheet melting and sterodynamic contributions. This leads for instance to sea-level rise that exceeds 0.5 m along most inhabited coastlines by 2100 under the RCP8.5 scenario (with respect to the period 1986-2005). Detail on the design of these projections and the related data are published in Le Cozannet et al. (2019b);

- high-end projections are derived by considering, for each sea-level component, the highest physically-based modelled estimate published in the literature. For the RPC8.5 scenario, for instance, we obtain a sea-level rise that exceeds 1.7 m for most of the European coastline by 2100 (with respect to the period 1986-2005). Detail on the design of these projections and the related data are published in Thiéblemont et al. (2019);

Note that there is no unique approach toward high-ends and low-ends. If we had relied on the expert elicitation of Bamber et al. (2019), our high-end projections would have been even higher. In this study, following the approach of IPCC (Oppenheimer et al., 2019), we rely on physical modelling outcomes only.

**4 Application of the framework**

**4.1 Analysis of past shoreline change**

Past shoreline changes are investigated first to ensure the validity of the modelling framework. Table 1 summarizes how uncertainties of each variables of Eq. 1 are defined (using either probability distributions or possibility distributions as introduced in Sect. 2) to model past shoreline changes in Sites 1 and 2, respectively. Over the historical period, mean sea level uncertainty for these two sites is assumed to be well represented by a normal probability distribution. For vertical ground motion (VGM), the sites that are investigated in our study have no statistically significant trends identified. Therefore, uncertainties due to VGM were prescribed as a centered normal distribution with a standard deviation of 2 mm/yr, as retrieved by the analysis of trends computed from the coastal permanent GNSS stations in the SONEL (Système d'Observation du Niveau des Eaux Littorales) database (Wöppelmann and Marcos, 2016). $Tx$ and $Lvar$ are also prescribed as normal probability distribution since they were derived assuming that errors of the linear regression are normally distributed (see section 2.b). Finally, as described in section 2.b, there is no consensus on the model to be used to project shoreline change in response to SLR. The design of the possibilistic distribution of the beach slope should therefore reflect this unknown by considering both the Bruun and the PCR model. The upper shoreface slopes are generally steeper than the nearshore slopes (e.g. 5-13% versus 1-2% in Aquitaine), applying the surrogate of the PCR model leads to reduced shoreline retreat estimates in comparison with the Bruun rule estimates (see section 2.b). Therefore, we defined the beach slope as an imprecise parameter which follows a possibilistic trapezoid distribution that span values ranging from the mildest records of the nearshore slope to steeper upper shoreface slopes. For Site 1, this leads to a core of the trapezoid in the range 1.2%-1.5% and a mildest slope of 1% (defining the origin of the support; see Table 1). For Site 2, the core of the trapezoid is in the range 2%-3.5% and the mildest slope is 1.5%. Finally, in absence of precise estimate of upper shoreface slopes sites 1 and 2, we use a uniform 10% slope as upper point of the trapezoid (Table 1).

| Variable | | Chosen uncertainty representation | Value | Data source |
|---|---|---|---|---|
| Past sea level changes | Site 1 | Probability - gaussian | 2.3 +/- 1 mm/year (1984-2014) | (Le Cozannet et al., 2019a) |
| | Site 2 | | 3.1 +/- 1 mm/year (1989-2019) | Marseille's tide gauge corrected from vertical ground motion. |
| Vertical ground motion | Site 1 | Probability - gaussian | 0 +/- 2 mm/year | Derived from (Wöppelmann and Marcos, 2016) |
| | Site 2 | | 0 +/- 2 mm/year | Derived from (Wöppelmann and Marcos, 2016) |
| tan β | Site 1 | Possibilistic – trapeze | [1%,1.2%,1.5%,10%] | Topographic & bathymetric survey |
| | Site 2 | | [1.5%,2%,3.5%,10%] | Topographic & bathymetric survey |

| | | | | |
|---|---|---|---|---|
| Lvar | Site 1 | Probability - gaussian | 0 +/- 7.3 m | Deduced from shoreline change observations (see Fig. 4a) |
| | Site 2 | | 0 +/- 5.0 m | Deduced from shoreline change observations (see Fig. 4b) |
| Tx | Site 1 | Probability - gaussian | 0.72 +/- 0.11 m/yr | Deduced from shoreline change observations (see Fig. 4a) |
| | Site 2 | | 0.50 +/- 0.07 m/yr | Deduced from shoreline change observations (see Fig. 4b) |

**Table 1: Chosen uncertainty representation (probabilistic or possibilistic) and data used to constrain input variables of equation 1. Note that Tx estimates appear slightly lower than trend estimates of Figure 4 as the past sea-level rise influence (using the Bruun rule) has been substracted.**

Fig. 5 shows the lower and upper probability bounds of past ΔS for the site #1 in Aquitaine and site #2 in Castellón. The results are derived from the uncertainty propagation scheme using 5000 random draw based on the uncertainty definition of each term of coastal impact model described in Table 1. For ease of comparison between the two sites, probability boxes are shown for a period of 10 years (gold) and 29-30 years (red) with respect to observational record references that are 2014 in Aquitaine and 2019 in Castellón.

For both sites, the gap between the lower and the upper bounds (i.e. the ambiguity) increases when moving increasingly backward in time (away from the reference year). This is expected and simply reflects the fact that uncertainty increases when exploring them further away from the reference date. In Aquitaine, the observed anomalous shoreline position for 1984 and 2004 are -14 m and -5 m, respectively (Fig. 4a). According to the associated p-boxes (Fig. 5a), the probability of exceedance of these two observed values are in the ranges 86%-
92%, and 65%-72%, hence well embedded within possibilistic bounds but also consistent with the fact that these observations appear to be well above (especially in 1984, upper ranges) the regression estimate (Fig. 4a). In Castellón, observed shoreline positions in 1990 (2009) are -13 (-8) m, which correspond to probability of exceedance in the ranges 73%-88% (29%-40%). Expanding our analysis to the entire profile of site 1, we found that 55% of the observations fall within the 25%-75% probability bounds and 100% within the 5%-95% confidence
limit. For site 2, we found that 78% of the observations fall within the 25-75% probability bounds and 96% within the 5%-95% confidence limit. This hindcast analysis hence suggests that our modelling framework is valid against observational historical records.

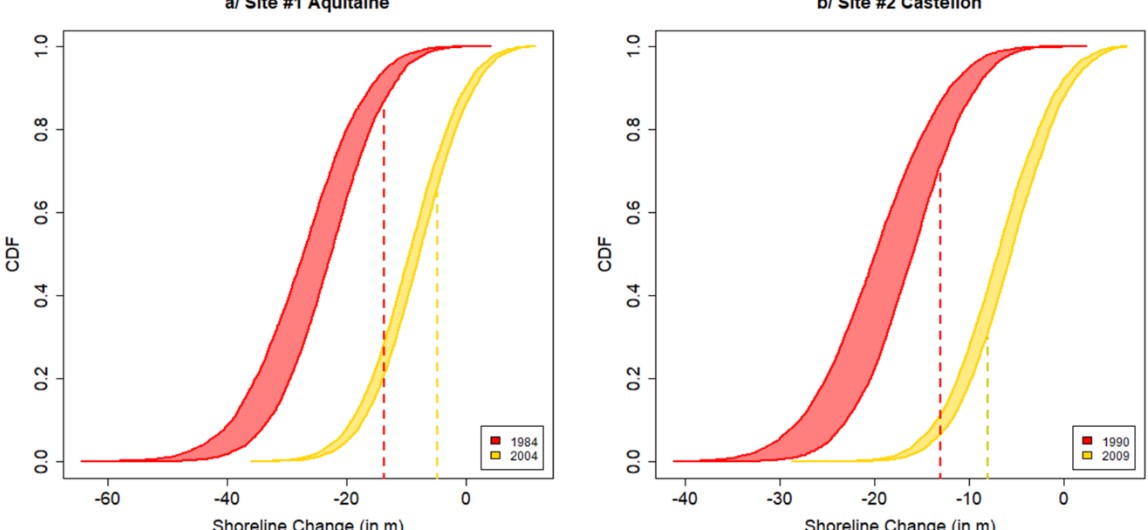

**Figure 5: Past shoreline change probability boxes in (a) Aquitaine in 1984 and 2004 and (b) Castellón in 1990 and 2009; i.e. distant by 10 years (yellow) and 30 years (red) from the observational reference, respectively. Vertical dashed line indicates observed values.**

## 4.2 Future projections of shoreline change

In contrast with the historical period for which observations of the mean sea-level are available and its uncertainty well quantified, projections of mean sea-level are deeply uncertain (see also introductory paragraph). This deep uncertainty source needs to be prescribed as input and, therefore, can no longer be considered as following a normal probabilistic distribution (as shown in Table 1). The relative sea-level change (RSLC) is defined as an imprecise input variable, which follows a trapezoidal possibility distribution, while all others inputs are taken identical to Table 1. We determine the RSLC possibility distribution for both time horizons 2050 and 2100, and three future climate change scenarios (RCP2.6, RCP4.5 and RCP8.5).

|  | RCP2.6 | | RCP4.5 | | RCP8.5 | |
|---|---|---|---|---|---|---|
|  | Aquitaine | Castellón | Aquitaine | Castellón | Aquitaine | Castellón |
| 2050 | [0.02,0.06, 0.22,0.31] | [0.07,0.10, 0.23,0.30] | [0.05,0.06, 0.24,0.39] | [0.09,0.11, 0.24,0.38] | [0.06,0.08, 0.27,0.50] | [0.09,0.12, 0.28,0.50] |
| 2100 | [0.08,0.12, 0.48,0.72] | [0.13,0.19, 0.52,0.72] | [0.19,0.20, 0.57,1.11] | [0.23,0.26, 0.62,1.16] | [0.37,0.39, 0.98,1.82] | [0.44,0.47, 1.03,1.83] |

**Table 2: RSLR (in m) projections in Aquitaine and Castellón for the RCP2.6, 4.5 and 8.5 scenarios in 2050 and 2100. RSLC projections are expressed as changes with respect to the year 2015. The first and fourth values in brackets correspond to RSLC estimates that define the support of the trapezoid (associated to a possibility degree of 0), and the second and third values in brackets correspond to RSLC estimates that define the core of the trapezoid (associated to a possibility degree of 1).**

Table 2 gives the values of RSLR (in m) for both time horizons and the three RCP scenarios used to construct the trapezoidal possibility distributions. The core of the trapezoid (possibility degree of 1) corresponds to the RSLR

likely-range as described in section 3.2. For instance, for the RCP8.5 projections in 2100 with respect to 2015 (and not 1986-2005 as in IPCC), we obtain likely ranges of 0.39-0.98 m in Aquitaine and 0.47-1.03 m in Castellón, which are both lower than the global mean sea level likely range of 0.55-1.04 m. This is consistent with the results

of Slangen et al. (2014) showing that North Atlantic and Mediterranean basins regional sea levels projections are beneath global mean sea level estimates. The boundaries of the support of the trapezoid, to which we assign a possibility degree of 0, correspond to the low-end (lower limit) and high-end (upper limit) RSLC estimates described in section 3.2.

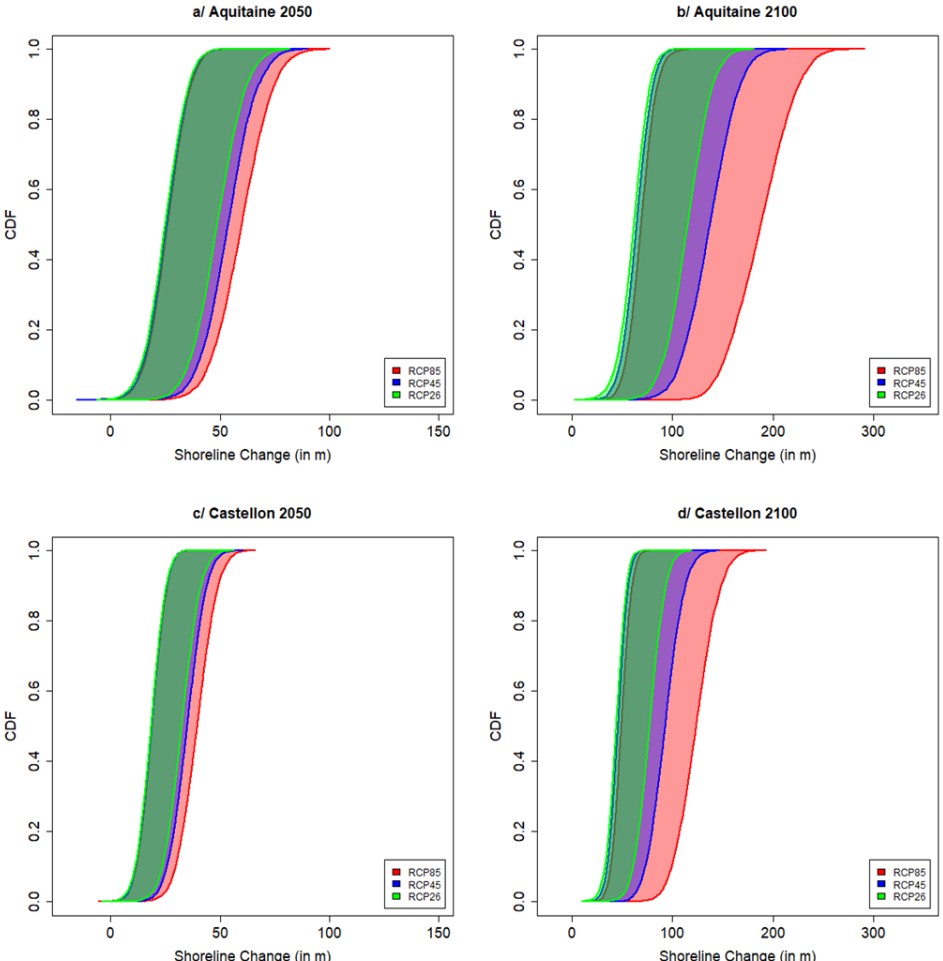

**Figure 6: Projected shoreline change probability boxes in (left) 2050 and (right) 2100 for (top) site #1 in Aquitaine and (bottom) site #2 in Castellón. Projections are shown for the (green) RCP2.6, (blue) RCP4.5 and (red) RCP8.5 scenarios. Ambiguity, low- and high-end corresponding values are given in Table 3.**

Fig. 6 shows the lower and upper probability bounds of ΔS projections for the sites 1 and 2 under three future RCP scenarios. In 2050, the difference in shoreline projections and their uncertainty between future scenarios is small

as shown by the median bounds which extends from 25m to 49m for the RCP2.6 and 26m and 60m for the RCP8.5 in Aquitaine (Fig. 6a). This result is consistent with the fact that SLR projections start to increasingly diverge after 2050 between the three future scenarios (Garner et al., 2018;Hinkel et al., 2019). In Castellón, small inter-scenario changes are also found (Fig. 6c) but lower median bounds are under those of the Aquitaine site; i.e. ~18 m for the Castellón site against ~25 m for the Aquitaine site. The upper median bound is also substantially more expanded

for the Aquitaine site (60 m) than the Castellón  site (40 m) when considering the RCP8.5 scenario. Therefore,

while scenario choice remains a modest source of uncertainty of shoreline projections by 2050, potential differences in nearshore slope and coastal impact models are already prominent. In 2100, ambiguity difference between RCP scenarios is strongly enhanced (see also Table 3). The upper uncertainty bound of the RCP8.5 scenario more than double those of the RCP2.6 scenario in both sites.

Table 3 provides the shoreline retreat thresholds of high-end and low-end scenarios associated with the probability boxes (and thresholds a and b) displayed on Fig. 6. Although defined arbitrarily, these two thresholds represent possible - but unlikely – "optimistic" and "pessimistic" future projections than can be considered as references to design minimum adaptation and maximum protection needs, respectively. In site 1 (site 2) in 2050, whatever the scenario, it appears that the shoreline could be retreating between ~24m (~16m) for a low-impact scenario and
more than 50 m (40 m) for a high-impact scenario. High-end values strongly increase in 2100, and could reach up to almost 200 m in site 1 and more than 130 m in site 2 under the RCP8.5 scenario. Under low-end scenarios, in 2100, 58 m and 37 m could still be lost in site 1 and site 2, respectively.

|  | Site 1 2050 | Site 1 2100 | Site 2 2050 | Site 2 2100 |
|---|---|---|---|---|
| Ambiguity [m] | **24**/**28**/**34** | **54**/**73**/**119** | **14**/**16**/**21** | **34**/**47**/**75** |
| Low-end [m] | **23**/**23**/**24** | **58**/**61**/**66** | **15**/**16**/**16** | **37**/**39**/**42** |
| High-end [m] | **52**/**57**/**63** | **120**/**144**/**196** | **35**/**38**/**43** | **85**/**100**/**132** |

**Table 3: Ambiguity, low- and high-end projected shoreline change thresholds [in m] in 2050 and 2100 for the site 1 in Aquitaine and the site 2 in Castellón. Green, blue and red numbers indicate thresholds are shown for the RCP2.6,**
**RCP4.5 and RCP8.5 scenarios, respectively.**

**4.3 Sensitivity analysis**

Shoreline change projections shown in Fig. 6 reveal that the uncertainty strongly amplifies with distant time horizons, in particular under high global warming scenarios. From a coastal planning perspective, such large uncertainties can be considered as unhelpful and not be used as such to support the decision making process
(Rohmer et al., 2019). In this case, it is particularly relevant to determine which uncertainty contributes the most to the total uncertainty in order to anticipate how foreseen improvements in the understanding of the physical system could reduce the uncertainty of projections. To this end, we performed a sensitivity analysis based on the pinching method (Tucker and Ferson, 2006). The pinching method consists of quantifying how the p-box changes if uncertain input parameters are pinched to a fixed value, i.e. assuming that the new knowledge context enables
to remove the corresponding epistemic uncertainty. The uncertain parameter leading to the maximum changes in the p-box is the one with the largest impact, i.e. the one that deserves further investigation in priority. Here, we pinch one parameter of Eq. (1) at a time and quantify the resulting effect on the ambiguity and high-end values.

Fig. 7 shows the results of the sensitivity analysis applied to site 1 for the RCP8.5 scenario in 2100. Note that this analysis has been extended to all scenarios and site 2 and revealed close results, leading to similar conclusions.
The figure reads e.g. as follows: assuming that the sea level off the Aquitaine coast would rise by 0.37 m in 2100 only (a very low estimate), the ambiguity (Fig. 7a) and high-end estimate (Fig. 7b) of shoreline change projection would both reduce by more than 50%. These results show that ambiguity and high-end estimate are primarily sensitive to uncertainty in SLR and beach slope. Ambiguity and high-end estimate in shoreline change projections

increase linearly with increasing SLR and decrease more abruptly (following an inverse function, consistent with Eq. (1)) with increasing beach slope. In comparison, the Tx and Lvar uncertainties have practically no effect on the ambiguity of shoreline change projections but show some influence on high-end estimates. The high-end shoreline change sensitivity to Tx and Lvar is also more pronounced in 2050 (not shown).

Interestingly, we note that the ambiguity increases when fixing SLR to high values (i.e. greater than 1.6 m, Fig. 7a). Intuitively, we expect the ambiguity to decrease when additional knowledge is provided; i.e. when the epistemic uncertainty is decreased. Yet, this holds only if the IRS-based randomly generated random intervals (see step 3 in Appendix A) are of lower widths given the fixed value. This is not always the case and depends on the characteristics (like the monotony) of the mathematical function optimized at step 3 (given the fixed value). Fig. S1 illustrates this effect by comparing the p-boxes for the case where the full SLR possibility distribution is considered against the case where the SLR value is fixed to 1.82 m. The p-box of the latter case shows an overall shift of the lower and upper CDF to higher values and a change in the width between the lower and upper CDF.

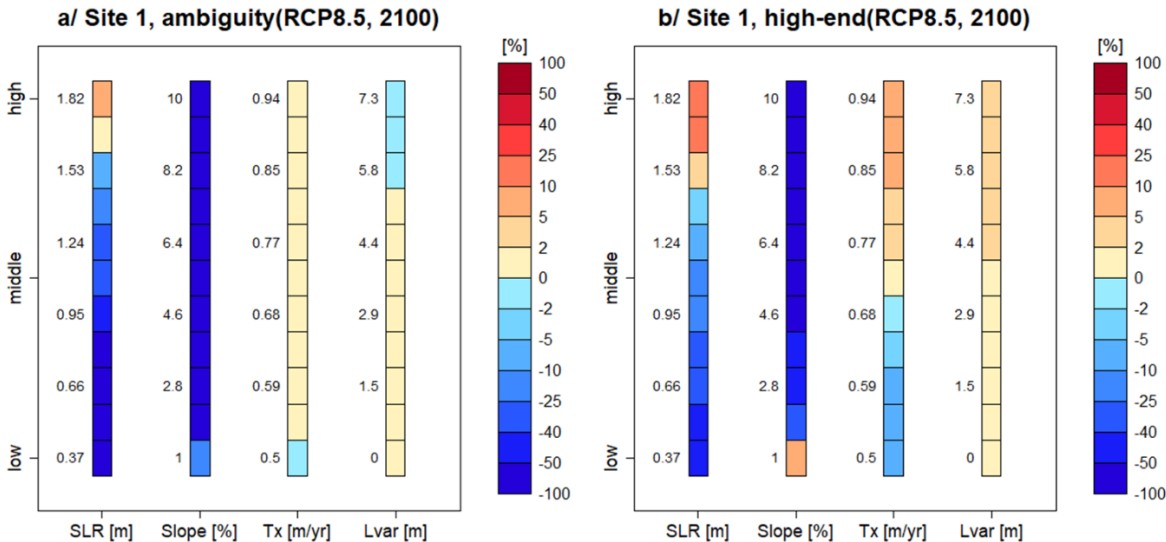

**Figure 7: Relative change (in %) of the p-box ambiguity (panel a) and high-end estimate (panel b) when the possibility (or probability) input distribution of one of the terms in Eq. (1) (i.e. SLR, Tx, …) is pinched to a fixed value. For each term, 11 values are pinched at a regular interval from the lower to the upper range of the possibility (or probability) distribution. These pinched values are specified on the left side of each bar plot. Site 1 is shown.**

This sensitivity analysis therefore suggests that improving both SLR projections and the understanding of their impact on shoreline could lead to a substantial reduction of uncertainty of future shoreline change. It should be emphasized that in the event that future SLR would not exceed the likely range (i.e. ~1 m), the ambiguity would be lowered by more than 50 %. Similarly, knowing exactly the nearshore slope contributes to drastically reduce the shoreline change uncertainty, in particular if this nearshore slope is steep (i.e. > 2%). Fixing the beach slope value in our simplified shoreline change equation implicitly suggests, though, that the coastal impact model is also well defined. The latter underlying assumption is however erroneous as reviewed previously (e.g. section 2.2). In the discussion, we explore in more details how shoreline change uncertainty is sensitive to the coastal impact model.

## 5 Discussion

### 5.1 Bruun vs. surrogate PCR model

By opting for a trapezoidal possibility distribution to represent the deep uncertainty on the nearshore slope as input of our shoreline change model, we recognize that the coastal impact is not well constrained since we assume together the Bruun rule and the surrogate PCR model within a single trapezoidal possibility distribution. We actually may wonder what would imply an improved knowledge of coastal impact models on shoreline change projections. In other words, how is the ambiguity affected if either the surrogate PCR model or the Bruun rule are excluded?

To address this question, we have changed the nearshore slope definition as input of our model. Results are shown on Fig. 8. To consider solely the Bruun model, beach slopes are defined as trapezoid considering the range 1.2%-1.5% for the core and 1%-1.6% for the support. Note that the 1.2% and 1.5% beach slopes correspond to the interval of foreshore slope from the dune toe to the depth of closure in Aquitaine (i.e. Bruun slopes). For the PCR model emulation, we adopted the approach of Le Cozannet et al. (2019a), where the slopes of the upper shoreface are substituted to the Bruun slopes. In site 1, slopes of the upper shoreface are comprised between 5% and 13%. Therefore, the PCR model was emulated by defining beach slopes as trapezoid considering the core 5.1%-12.9% and the support 5%-13%.

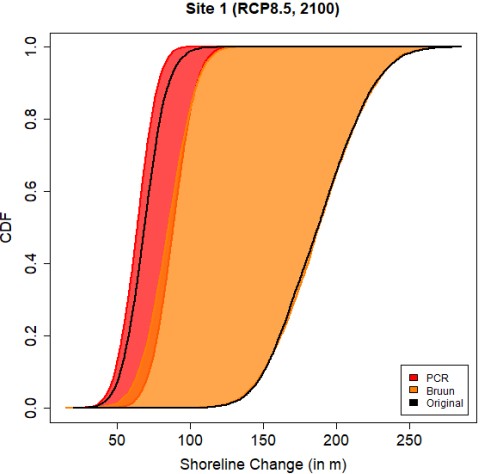

**Figure 8: Projected shoreline change p-box in 2100 for site 1 under the RCP8.5 scenario. The reference model is displayed with the black envelop and the modified models following the PCR model emulation, Bruun rule, are displayed in red, orange, respectively.**

On Fig. 8, the PCR model emulation (red) and Bruun model (orange) realizations are compared to the reference model (black p-box envelop). Our results reveal that the Bruun model fits nicely the upper bound of the reference model and encompass most of the ambiguity. In contrast, the p-box built from the PCR emulation model overlaps the reference model in its lower bounds and has an area four times smaller than for the Bruun model. Therefore, considering the PCR model leads to a strong reduction of the uncertainty of SLR-induced shoreline change but also to a sharp decrease of projected coastline retreat. This is due to the fact that the SLR-induced shoreline change is proportional to the inverse of the beach slope, which varies weakly on the range of beach slopes 5%-13%.

Conversely, the Bruun model exacerbates shoreline change ambiguity and shoreline change sensitivity to SLR uncertainties.

### 5.2 Considering anthropization

Along the Castellón coastal stretch, most sectors have been affected by human intervention. This implies that great caution is needed when applying our simple shoreline change model for this area. For instance, in Almardà (South of Chilches), beach nourishments have been carried out over the 1995-1998 and 2010-2013 periods, resulting in an overall beach accretion of 1.5 m/year over the 1989-2019 period as shown on Fig. 9a. Outside beach nourishment periods though, shoreline retreat is observed as revealed by positive trend displayed in red. Although

our shoreline change model does not explicitly include past anthropogenic influences, effects such as beach nourishment can be implicitly accounted for in the $Tx$ and $Lvar$ terms. For instance, for Almadarà (Fig. 9a), the $Tx$ is negative (i.e. beach accretion) due to beach nourishment. Therefore, shoreline change projections made for this site would assume that beach nourishment will be pursued in the future at the same rates and frequency. In such a case, our projections show that by 2100 and even under the RCP8.5, shoreline is expected to further progress

toward the sea, with a very large uncertainty though as revealed by the black p-box Fig. 9b.

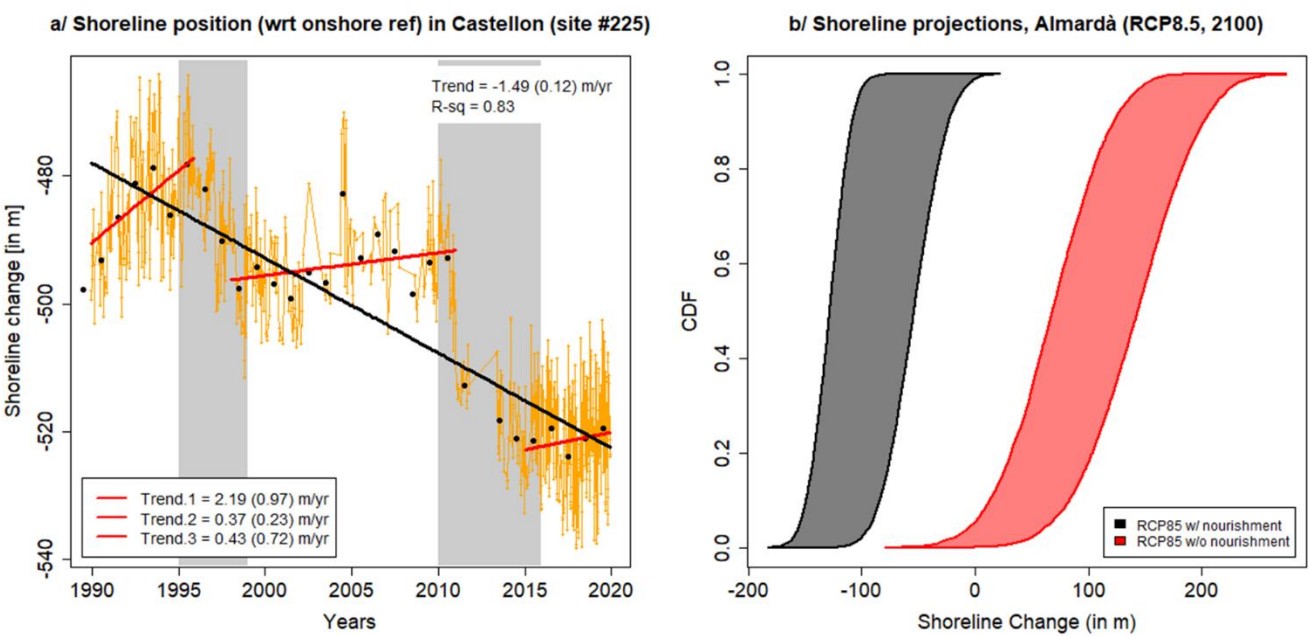

**Figure 9: (a) Observed shoreline change evolution in Almardà. The black and red lines indicate trends computed with and without, respectively, the periods of beach nourishment, which are displayed in gray. (b) Probability boxes of projected shoreline change by 2100 under the RCP8.5 scenarios with (black) and without (red) including beach**

**nourishment.**

Assuming that beach nourishment will continue is however strongly uncertain and should be avoided. In this regard, we derived the $Tx$ term by relying only on periods outside beach nourishment shown by the red segments. This leads to a weighted mean $Tx$ of 0.83 m/year. The resulting projections in 2100 under the RCP8.5 scenario are shown by the red p-box, which in this case clearly indicates that in absence of future beach nourishment, the

shoreline is projected to retreat in face of sea-level rise. The ambiguity remains very similar, indicating that

accounting for beach nourishment simply translates the p-box. Nonetheless, we note that when the nourishment is not included, the p-box is more tilted, which is due to the higher standard error associated to the $Tx$ term.

### 5.3 Advantages of extra-probabilistic approaches

Here, we discuss the advantage of the use of possibilities in comparison to e.g. a modelling framework that would be fully probabilistic. To illustrate this, we re-calculate shoreline change projections with Eq. (1) in Aquitaine (site 1) in 2100 but assuming that $\Delta RSLC$ and $\tan\beta$ follow normal distributions. We consider the RCP8.5 scenario with $\Delta RSLC$ defined as $0.69\text{m} \pm 0.24\text{m}$ and the Bruun rule with $\tan\beta$ defined as $1.35\% \pm 0.15\%$. The resulting shoreline change projections are normally distributed with 5th and 95th percentiles of 71m and 152m, respectively. Within a probabilistic approach, these left and right tails can be reasonably associated to low and high-end projections. The comparison with extra-probabilistic low and high-end projections in Table 3 (i.e. 66m and 196m, respectively) shows substantial differences, and in particular that high-end values obtained within the probabilistic theory are much lower. More importantly, we found that the high-end projection obtained with the possibilistic framework is not even achievable under the probabilistic model built here; hence indicating that the probability-based high-end scenario is too optimistic in the sense that it fails to reflect deep uncertainty. One should thus design dedicated (and separated) high-end scenarios to explore such projections that may appear arbitrary.

Another strength of the extra-probabilistic framework is its flexibility with respect to the available expert data, which allows easily fusing different low- and high- end scenarios. In this study, we accounted for the deep uncertainty in future SLR by designing RSLC projections that follow a trapezoid possibility function and selecting a set of low- and high-end estimates to bound the support of the trapezoid. As mentioned in section 3.2, there is yet no unique approach to design low- and high-end projections. Possibility functions can therefore be adapted to encompass multiple high-end estimates. For instance Le Cozannet et al. (2017) translated experts opinion on future Antarctica contribution into three different possible upper bounds for 2100 sea-level rise. These estimates were then aggregated into a single stair-like input function where the three high-end scenarios were assigned with various degrees of possibility. Applying this aggregated possibility distribution in our case would result in similar ambiguity estimates but with an increase of the p-box's upper tail (for percentile superior to 90%) up to values of 500m (by 2100 for Aquitaine, see Fig. S2).

Finally, the problem of model uncertainty related to the use of the Bruun or the surrogate PCR model provides a good illustration of how the quantified measure of ambiguity in the projection can de decomposed. The use of possibilities allows making very transparent the ambiguity thanks to the p-boxes graphical representation. This has also the advantage of showing how future progress in the system knowledge may contribute reducing deep uncertainty. From a decision-making perspective, the extra-probabilistic approach thus allows a transparent and exhaustive consideration of uncertainties. One should nonetheless bear in mind that in case where knowledge uncertainty becomes very prominent and requires an extensive use of possibility distribution as input, the ambiguity in the outcome may be considered by end-users as too large to be informative and useful.

### 6 Conclusion

The approach presented in this paper provides a framework for assessing deep uncertainties in shoreline change projections. This framework is versatile since the definition of input variables and their distribution can be adapted easily to the characteristics of a local site, its data coverage and the degree of knowledge of hydrosedimentary

processes acting locally. Furthermore, this extra-probabilistic approach that we here apply to an empirical shoreline evolution model can be actually replicated to any of the available models of shoreline evolution (Montaño et al., 2020).

In our approach, residual uncertainties that have not been integrated quantitatively still remain. For example, the Bruun rule and the PCR models are not the only plausible models for shoreline change reconstructions. Similarly, our high-end sea-level rise estimates might be exceeded by 2050 according to recent expert elicitation of the future contribution of Greenland and Antarctica ice-sheets to sea-level rise (Bamber et al., 2019). The approach consisting in summing up the different modes of variability of shoreline change can also be challenged on the ground. For example, coastal defenses may limit the potential retreat of shorelines in other areas. Finally, future adaptation is unknown and could limit or favor coastal erosion and shoreline changes.

Despite these limitations, our approach is potentially useful to determine to which extent reducing our uncertainties on e.g. future sea-level rise or coastal impact models can help improving the precision of future shoreline change projections. For example, we have shown that if sea-level rise does not exceed 1m, shoreline change uncertainties will be reduced significantly. This could be achieved through an ambitious climate mitigation policy and improved knowledge on ice-sheets. While there remain the issue of the long term commitment to sea-level rise (Clark et al., 2016), reducing this source of deep uncertainties would grant more time for coastal adaptation.

**Appendix A**

**The Independent Random Sampling (IRS) algorithm**

Consider $k$ random input variables $X_i$ (i=1,...,$k$), each of them associated to a cumulative probability distribution $F$, and $n$-$k$ imprecise input variables $X_i$ (i=$k+1$,…,$n$), each of them associated to a possibility distribution $\pi$. In this situation, the IRS procedure holds as follows:

- Step 1. Randomly generate from uniform probability distributions, $m$ vectors of size $n$: $\{\alpha_i\}$, i=1,…,$n$, such that $0\leq\alpha_i\leq1$. For each realization:

- Step 2. Generate $k$ values for the random input variables by using the inverse function of $F_i$: $x_i = F_i^{-1}(\alpha_i)$, i=1,...,$k$ and sample $n$-$k$ intervals $I_i$ corresponding to the cuts of the possibility distributions (as defined in Sect. 2.1 and illustrated in Fig. 1) with level of confidence $1$-$\alpha_i$, i=$k+1$,…,$n$;

- Step 3. Evaluate the interval $[\underline{h};\overline{h}]$ defined by the lower and upper bounds associated to the model output $h$ (in our case, the shoreline change) using the impact assessment model $f$ as follows:

$$\underline{h} = \inf_I\big(f(x_1; …; x_k; I_{k+1}; …; I_n)\big) \ ; \ \overline{h} = \sup_I\big(f(x_1; …; x_k; I_{k+1}; …; I_n)\big)$$

Fig. A1 schematically depicts the main steps of the propagation procedure considering a random and an imprecise variable. The output of the whole procedure then takes the form of $m$ random intervals $[\underline{h};\overline{h}]$, with $k=1,...,m$. This information can be summarized within the formal framework of the evidence theory (Dempster, 1967; Shafer, 1976) as proposed by Baudrit et al. (2005) to bound the exceedance probability associated to the event "$h\geq t_h$" with $t_h$ a given threshold. The result then takes the form of the probability-boxes as depicted in e.g. Fig. 5.


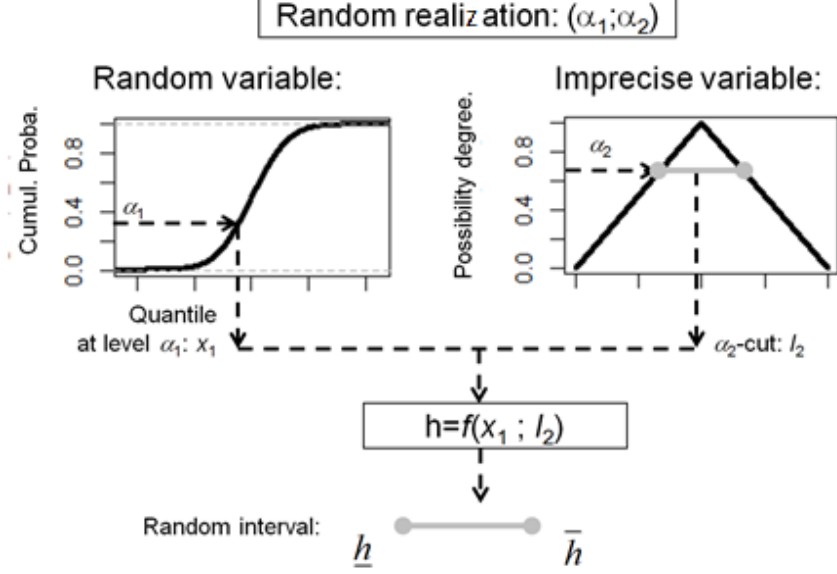

Figure A1. Overview of the main steps for joint propagation of possibility and probability distributions. Adapted from Rohmer and Verdel (2014).

**Data availability**

AR5 projections can be downloaded from the Integrated Climate Data Center at the University of Hamburg (https://icdc.cen.uni-hamburg.de/en/ar5-slr.html, University of Hamburg, 2021). Other projections (i.e. low-end, high-end projections) used in this study are available at https://sealevelrise.brgm.fr/. Other data such as shoreline observations used in this paper will be made available with the code.

**Code availability**

HYRISK software, used to design and propagate jointly probability and possibility distributions, is a publicly available CRAN R package (https://cran.r-project.org/web/packages/HYRISK/index.html). R code and data needed to reproduce simulations, shoreline reconstructions and projections, and related figures for each case study will be provided in a dedicated repository.

**Author contribution**

RT designed the sea-level projections, performed the data analysis and the simulations. GL had the original idea of the study, contributed to the design and to the interpretation of the simulations. JR developed the HYRISK software used to perform the simulations and participated to their interpretation. AT, MAC and IJL retrieved the shoreline data in Castellón and contributed to interpret their projections. RT prepared the manuscript with
contributions from all co-authors.

**Competing interests**

The authors declare that they have no conflict of interest.

**Acknowledgements**

This research was funded by the BRGM, IHCantabria and the ERA4CS-ECLISEA project (grant number: 690462),
and takes part in the COasTAUD framework. We thank Mark Carson for making available the ICDC sea-level
data and the modelling groups that participated in the CMIP5 for producing their model output. We thank also
SONEL and PSMSL services for providing tides gauges and GPS land motion records. We thank Bruno Castelle
for providing shoreline change historical evolution dataset in Aquitaine. We thank the two anonymous reviewers
for their constructive comments on our work.

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
