# Peer review of "Deep uncertainties in shoreline change projections: an extraprobabilistic approach applied to sandy beaches"

_Natural Hazards and Earth System Sciences, 2020_

## Author Comment (AC1)

**Reviewer #1**

This paper examines the extent of deep uncertainty in studies on projections of shoreline changes. First of all, I find elegant the definition of deep uncertainty based based on the concept of "possibility" in an "extra-statistics" context.
This could be indeed an interesting tool to assess more correctly the variability of predictions, when we have a reasonable range for the epistemic uncertainty.
The paper is well written and well structured. The explanation of what possibility is and of the model(s) used for shoreline predictions and their parameters is clear.
I believe a few lines would be useful, summerizing how the montecarlo over the "possibility", possibly with references. Is a possibility distribution treated mathematically like a probability distribution? Or is it used to select a CDF?
Apart from this request of clarification, I have only minor comments.

We thank the reviewer for her/his insightful comments that, we believe, will contribute to improve the clarity of our manuscript. Please find below our responses (in blue) and how the manuscript will be revised (preceded by an arrow).

I believe a few lines would be useful, summerizing how the montecarlo over the "possibility", possibly with references. Is a possibility distribution treated mathematically like a probability distribution? Or is it used to select a CDF?

The procedure for jointly propagating probability and possibility distributions builds on the combination of random sampling of the inverse of the cumulative probability distribution functions for random parameters and of the $\alpha$-cuts (intervals associated to a level of confidence of 1-$\alpha$) from the possibility distributions using the Independent Random Sampling (IRS) algorithm of Baudrit et al. (2007). The IRS algorithm is described below.

Consider $k$ random input variables $X_i$ (i=1,...,$k$), each of them associated to a cumulative probability distribution $F$, and $n$-$k$ imprecise input variables $X_i$ (i=$k$+1,...,$n$), each of them associated to a possibility distribution $\pi$. In this situation, the IRS procedure holds as follows:

- Step 1. Randomly generate from uniform probability distributions, $m$ vectors of size $n$: $\{\alpha_i\}$, i=1,...,$n$, such that $0 \leq \alpha_i \leq 1$. For each realization:
- Step 2. Generate $k$ values for the random input variables by using the inverse function of $F_i$: $x_i = F_i^{-1}(\alpha_i)$, i=1,...,$k$ and sample $n$-$k$ intervals $I_i$ corresponding to the cuts of the possibility distributions (as defined in Sect. 2.1 and illustrated in Fig. 1) with level of confidence 1-$\alpha_i$, i=$k$+1,...,$n$;
- Step 3. Evaluate the interval $[\underline{h};\overline{h}]$ defined by the lower and upper bounds associated to the model output $h$ (in our case, the shoreline change) using the impact assessment model $f$ as follows:

$$\underline{h} = \inf_I \big(f(x_1; ...; x_k; I_{k+1}; ...; I_n)\big) ; \quad \overline{h} = \sup_I \big(f(x_1; ...; x_k; I_{k+1}; ...; I_n)\big)$$

Figure R1 schematically depicts the main steps of the propagation procedure considering a random and an imprecise variable. The output of the whole procedure then takes the form of $m$ random intervals $[\underline{h};\overline{h}]$, with $k$=1,...,$m$. This information can be summarized within the formal framework of the evidence theory (Dempster, 1967; Shafer, 1976) as proposed by Baudrit et al. (2006) to bound the exceedance probability associated to the event "$h \geq t_h$" with $t_h$ a given threshold. The result then takes the form of the probability-boxes as depicted in Fig. 5.

[Figure]

*Figure R1.* Overview of the main steps for joint propagation of possibility and probability distributions.

> ⇨ The manuscript will be revised by adding few lines of clarification about IRS procedure near L200-207 and by adding an Appendix at the end of the manuscript describing the IRS algorithm.

- formula 1: the angle is missing after tan. I would suggest to call the angle "beta" as "alpha" can be confused with the confidence level used earlier, and beta this is the symbol generally used in the Bruun formula.

Thank you for noticing.
> ⇨ We'll change α for β in the revised version of the manuscript and correct the formula.

- line 156, a brief explanation of what nx, Tx and lvar are should be given here, mentioning that the terms will be explained later in more detail.

We agree that the term *n Tx* can be better defined.

> ⇨ We'll now write: " *Tx* is the linear trend of shoreline changes over multi-decadal shoreline change and *n* the number of years relative to the baseline"

We do not want however to enter too much in details here since in the subsequent paragraph (starting at line 164), we provide additional explanations on these terms, mentioning that they correspond to evolutions over different timescales, and are not associated to particular physical processes. We remind also that relying on observations to assess trends and modes of variability of shoreline change is common practice in operational shoreline change management.

- line 157, substitute "in the following" with "below".

> ⇨ This will be corrected in the revised version of the manuscript.

- figure 2: I would appreciate a brief summery on how the hybrid montecarlo works.

Please refer to the 1st comment above.

⇨ The manuscript will be revised by adding few lines to clarify the IRS procedure near L200-207 and by adding an Appendix at the end of the manuscript describing the IRS algorithm.

- line 173-174: is lvar an uncertainty term, with and unknown sign? It would look like this, as it is computed as a standard deviation. This should be made more clear, for example writing +/- lvar in the formula.

In fact, Lvar is a particular realization of a random variable, and not a random variable itself. Although we prefer keeping the formula as it is, we agree that this deserves being clarified in the paper.

⇨ We'll include a sentence clarifying the issue of the sign just after describing how Lvar is derived.

- line 277: the acronym GNSS should be defined.

GNSS stands for Global Navigation Satellite System.
⇨ This will be added in the revised version of the manuscript.

- line 308: the acronym SONEL shoudl be defined.

SONEL stands for "Système d'Observation du Niveau des Eaux Littorales".
⇨ This will be added in the revised version of the manuscript.

- table 1: I would suggest to indicate also the possibilistic choice of the model here.

Our apologies but we are not sure to understand well the suggestion. Actually the choice of the model is already transformed into a possibilistic framework through the possibility distribution built for the tan α parameter.

⇨ We can propose to clarify this by adding a note to the tan α row in Table 1.

- line 240 and elsewhere: check what sign you use to indicate erosion/accretion. Here and in figure 4, a positive change is erosion. But looking at figure 5, it looks that the reverse convetion is used (a negative shoreline change where you have erosion).

Thank you for this comment. Assuming an observational reference near present (~2015 for Aquitaine and 2020 for Castellon), Figure 5 shows that when moving backward in time, we indeed have – relatively - an accretion in the past compared to the reference period (so increasingly negative values when moving further in the past). In contrast, when moving forward in the future, both sites are eroding (so increasing positive values when moving further in the future). Thus, in principle, the same convention appears to be used everywhere. We recognize however that this can be confusing and will make sure that the same convention is used throughout the revised manuscript.

⇨ Just before presenting any result, we'll include a statement clarifying the convention we use such as: "Thereinafter, positive and negative values represent erosion and accretion, respectively, with respect to the baseline (2015 for Aquitaine and 2020 for Castellon)"

- figure 6: I would suggest to add lines to identify the extent/position of the ambiguity and of the high/low ends.

As suggested, we re-designed Figure 6b (i.e. Aquitaine, 2100) in order to visualize the extent of the ambiguity and the position of high and low end values. The resulting Figure R2 appears heavier to us and we have the feeling that it does not bring real additional information since high/low-end and ambiguity values are already quoted in Table 3. Note also that Figure 6b is the panel of Figure 6 that offers the largest space to visually separate the vertical lines and bars with the suggested design; Figure 6a and c will look more loaded. An alternative could be to suggest to the journal a page layout joining Figure 6 and Table 3 on the same page. If the choosing the page layout is not possible, another alternative could be to include a sentence with additional information on these values in Figure caption.

[Figure]

**Figure R2.** *Alternative design to Figure 6b, highlighting (i) the ambiguity by horizontal bars in the upper part and (ii) low/high-end values by vertical lines.*

- figure 7: Is it possible that fixing some param values the ambiguity increases? Maybe a line explaining this would be useful.

Thank you for noticing. Intuitively, we expect the ambiguity to decrease when we add knowledge i.e. when the epistemic uncertainty is decreased; for instance when SLR is fixed to a constant value. Yet, this is only valid if the IRS-based randomly generated random intervals (see step 3 in the IRS procedure; first comment) are of lower widths given the fixed value. This is not always the case and depends on how the characteristics of the mathematical function (i.e. given the fixed value) are optimized at step 3. In our case, Figure 7 shows that fixing the future SLR to very high values (e.g. 1.82 m) leads to an increase of the ambiguity and high-end values of a few percent. To illustrate this effect more clearly, Figure R3 below compares the resulting possibility box when considering either (a) the full SLR possibility distribution or (b) only a high-end fixed SLR value of 1.82 m. It shows that the shape of the possibility box is modified with an overall shift of the lower and upper CDF to higher values and a change in the width between the lower and upper CDF (note that this change also varies with quantiles).

[Figure]

***Figure R3.*** *Projected shoreline change probability boxes in 2100 in Aquitaine under the RCP8.5 scenario when (a) considering the full SLR possibility distribution or (b) fixing the SLR to a high-end value (i.e. 1.82 m).*

⇨ We will add few elements of explanation in the revised version of the manuscript where Figure 7 is discussed (L409-418) and will add the Figure R3 above as supplementary material.

- figure 8: the figure is truncated, the x axis is missing.

Thank you for noticing.

⇨ Figure 8 will be adjusted in the revised version of the manuscript.

**References:**

Baudrit, C., Guyonnet, D., and Dubois, D., 2006, "Post-processing the hybrid method for addressing uncertainty in risk assessments," Journal of Environmental Engineering, 131, pp. 1750-1754.

Baudrit C, Guyonnet D, Dubois D (2007) Joint propagation of variability and imprecision in assessing the risk of groundwater contamination. Journal of contaminant hydrology, 93(1):72-84.

Dempster, A. P., 1967, "Upper and lower probabilities induced by a multivalued mapping," Annals of Mathematical Statistics, 38, pp. 325–339.

Shafer, G., 1976, A Mathematical Theory of Evidence. Princeton University Press

---

## Author Response (AR1)

**Reviewer #1**

This paper examines the extent of deep uncertainty in studies on projections of shoreline changes. First of all, I find elegant the definition of deep uncertainty based on the concept of "possibility" in an "extra-statistics" context.
This could be indeed an interesting tool to assess more correctly the variability of predictions, when we have a reasonable range for the epistemic uncertainty.
The paper is well written and well structured. The explanation of what possibility is and of the model(s) used for shoreline predictions and their parameters is clear.
I believe a few lines would be useful, summerizing how the montecarlo over the "possibility", possibly with references. Is a possibility distribution treated mathematically like a probability distribution? Or is it used to select a CDF?
Apart from this request of clarification, I have only minor comments.

We thank the reviewer for her/his insightful comments that, we believe, will contribute to improve the clarity of our manuscript. Please find below our responses (in blue) and how the manuscript has been revised (preceded by an arrow). Note that lines numbering refer to the track changed manuscript.

I believe a few lines would be useful, summerizing how the montecarlo over the "possibility", possibly with references. Is a possibility distribution treated mathematically like a probability distribution? Or is it used to select a CDF?

The procedure for jointly propagating probability and possibility distributions builds on the combination of random sampling of the inverse of the cumulative probability distribution functions for random parameters and of the $\alpha$-cuts (intervals associated to a level of confidence of 1-$\alpha$) from the possibility distributions using the Independent Random Sampling (IRS) algorithm of Baudrit et al. (2007). The IRS algorithm is described below.

Consider $k$ random input variables $X_i$ (i=1,...,$k$), each of them associated to a cumulative probability distribution $F$, and $n$-$k$ imprecise input variables $X_i$ (i=$k$+1,…,$n$), each of them associated to a possibility distribution $\pi$. In this situation, the IRS procedure holds as follows:

- Step 1. Randomly generate from uniform probability distributions, $m$ vectors of size $n$: {$\alpha_i$}, i=1,…,$n$, such that 0≤$\alpha_i$≤1. For each realization:
- Step 2. Generate $k$ values for the random input variables by using the inverse function of $F_i$: $x_i = F_i^{-1}(\alpha_i)$, i=1,...,$k$ and sample $n$-$k$ intervals $I_i$ corresponding to the cuts of the possibility distributions (as defined in Sect. 2.1 and illustrated in Fig. 1) with level of confidence 1-$\alpha_i$, i=$k$+1,…,$n$;
- Step 3. Evaluate the interval [$\underline{h}$;$\overline{h}$] defined by the lower and upper bounds associated to the model output $h$ (in our case, the shoreline change) using the impact assessment model $f$ as follows:

$$\underline{h} = \inf_I\left(f(x_1; ...; x_k; I_{k+1}; ...; I_n)\right); \ \overline{h} = \sup_I\left(f(x_1; ...; x_k; I_{k+1}; ...; I_n)\right)$$

Figure R1 schematically depicts the main steps of the propagation procedure considering a random and an imprecise variable. The output of the whole procedure then takes the form of $m$ random intervals [$\underline{h}$;$\overline{h}$], with $k$=1,...,$m$. This information can be summarized within the formal framework of the evidence theory (Dempster, 1967; Shafer, 1976) as proposed by Baudrit et al.

(2006) to bound the exceedance probability associated to the event "$h \geq t_h$" with $t_h$ a given threshold. The result then takes the form of the probability-boxes as depicted in Figs. 5 and 6.

[Figure]

**Figure R1.** *Overview of the main steps for joint propagation of possibility and probability distributions.* Adapted from Rohmer and Verdel (2014).

⇨ We added a sentence (L210-213) to describe the hybrid Monte-Carlo principle briefly and then added Appendix A (L579-600) devoted to the description of the IRS algorithm.

- formula 1: the angle is missing after tan. I would suggest to call the angle "beta" as "alpha" can be confused with the confidence level used earlier, and beta this is the symbol generally used in the Bruun formula.

Thank you for noticing.

⇨ We changed α for β everywhere in the revised version of the manuscript (including Figure 2) and corrected the formula.

- line 156, a brief explanation of what nx, Tx and lvar are should be given here, mentioning that the terms will be explained later in more detail.

We agree that the term *n Tx* can be better defined.

⇨ We have now clarified: " *Tx* is the linear trend of shoreline changes over multi-decadal timescales and *n* the number of years relative to the baseline" (L156-157.)

We do not want however to enter too much in details here since in the subsequent paragraph (starting at line 166), we provide additional explanations on these terms, mentioning that they correspond to evolutions over different timescales, and are not associated to particular physical processes. We remind also that relying on observations to assess trends and modes of variability of shoreline change is common practice in operational shoreline change management.

- line 157, substitute "in the following" with "below".

⇨ This has been corrected (L160).

- figure 2: I would appreciate a brief summery on how the hybrid montecarlo works.

Please see our answer to the 1st comment above.

- line 173-174: is lvar an uncertainty term, with and unknown sign? It would look like this, as it is computed as a standard deviation. This should be made more clear, for example writing +/- lvar in the formula.

Although we prefer keeping the formula as it is, we agree that this deserves being clarified in the paper.

⇨ We modified Table 1 to make clear how Lvar is defined as input distribution. We have included a sentence clarifying the issue of the sign just after describing how Lvar is derived: "typically, Lvar would quantify how the shoreline can depart from a mean position due to e.g., seasonal cycles or the chronological sequence of storms and calm period" (L158-159)

- line 277: the acronym GNSS should be defined.

GNSS stands for Global Navigation Satellite System.
⇨ Clarified (L285).

- line 308: the acronym SONEL shoudl be defined.

SONEL stands for "Système d'Observation du Niveau des Eaux Littorales".
⇨ Clarified (L344).

- table 1: I would suggest to indicate also the possibilistic choice of the model here.

We understand that the reviewer highlights that a choice has been made to select a particular uncertainty representation (e.g., probabilistic or possibilistic distribution).

⇨ We have now clarified this in the caption and in the table (L360).

We recall also that, in Section 4.1, it was stated that the uncertainty in the model choice was reflecting through the choice of tan(β): " Finally, as described in section 2.b, there is no consensus on the model to be used to project shoreline change in response to SLR. The design of the possibilistic distribution of the beach slope should therefore reflect this unknown by considering both the Bruun and the PCR model. The upper shoreface slopes are generally steeper than the nearshore slopes (e.g. 5-13% versus 1-2% in Aquitaine), applying the surrogate of the PCR model leads to reduced shoreline retreat estimates in comparison with the Bruun rule estimates (see section 2.b). Therefore, we defined the beach slope as an imprecise parameter which follows a possibilistic trapezoid distribution that span values ranging from the mildest records of the nearshore slope to steeper upper shoreface slopes."

- line 240 and elsewhere: check what sign you use to indicate erosion/accretion. Here and in figure 4, a positive change is erosion. But looking at figure 5, it looks that the reverse convetion is used (a negative shoreline change where you have erosion).

Thank you for this comment. Assuming an observational reference near present (~2015 for Aquitaine and 2020 for Castellon), Figure 5 shows that when moving backward in time, we indeed have – relatively - an accretion in the past compared to the reference period (so increasingly negative values when moving further in the past). In contrast, when moving forward in the future, both sites are eroding (so increasing positive values when moving further in the future). Thus, in principle, the same convention appears to be used everywhere. We recognize however that this can be confusing and will make sure that the same convention is used throughout the revised manuscript.

⇨ We have included a general statement upfront that clarifies the convention we use (L235-236).

- figure 6: I would suggest to add lines to identify the extent/position of the ambiguity and of the high/low ends.

As suggested, we re-designed Figure 6b (i.e. Aquitaine, 2100) in order to visualize the extent of the ambiguity and the position of high and low end values. The resulting Figure R2 appears heavier to us and we have the feeling that it does not bring real additional information since high/low-end and ambiguity values are already quoted in Table 3. Note also that Figure 6b is the panel of Figure 6 that offers the largest space to visually separate the vertical lines and bars with the suggested design; Figure 6a and c will look more loaded. An alternative could be to suggest to the journal a page layout joining Figure 6 and Table 3 on the same page. If the choosing the page layout is not possible, another alternative could be to include a sentence with additional information on these values in Figure caption.

[Figure]

***Figure R2.*** *Alternative design to Figure 6b, highlighting (i) the ambiguity by horizontal bars in the upper part and (ii) low/high-end values by vertical lines.*

⇨ For now, we added a reference to Table 3 in the caption of Figure 6. We'll ask if it is feasible to display Table 3 and and Figure 6 on the same page.

- figure 7: Is it possible that fixing some param values the ambiguity increases? Maybe a line explaining this would be useful.

Thank you for noticing. Intuitively, we expect the ambiguity to decrease when we add knowledge i.e. when the epistemic uncertainty is decreased; for instance when SLR is fixed to a constant value. Yet, this is only valid if the IRS-based randomly generated random intervals (see step 3 in the IRS procedure; first comment) are of lower widths given the fixed value. This is not always the case and depends on the characteristics (like the monotony) of the

mathematical function optimized at step 3 (given the fixed value). In our case, Figure 7 shows that fixing the future SLR to very high values (e.g. 1.82 m) leads to an increase of the ambiguity and high-end values of a few percent. To illustrate this effect more clearly, Figure R3 below compares the resulting possibility box when considering either (a) the full SLR possibility distribution or (b) only a high-end fixed SLR value of 1.82 m. It shows that the shape of the possibility box is modified with an overall shift of the lower and upper CDF but with different magnitudes; the shift of the upper CDF being larger for the percentile range over 0.4-0.6, hence implying an increase of the ambiguity. Note that if the ambiguity has been defined as the width between the 90% percentile (instead of the medians) of the upper and lower CDF, the pinching would have led to a decrease of ambiguity.

[Figure]

*Figure R3. Projected shoreline change probability boxes in 2100 in Aquitaine under the RCP8.5 scenario when (a) considering the full SLR possibility distribution or (b) fixing the SLR to a high-end value (i.e. 1.82 m).*

⇨ We added a paragraph to comment the increase in ambiguity (L459-466) and Figure above is now added in the supplementary material as Figure S1.

- figure 8: the figure is truncated, the x axis is missing.

Thank you for noticing.

⇨ Corrected

**References:**

Baudrit, C., Guyonnet, D., and Dubois, D., 2006, "Post-processing the hybrid method for addressing uncertainty in risk assessments," Journal of Environmental Engineering, 131, pp. 1750-1754.

Baudrit C, Guyonnet D, Dubois D (2007) Joint propagation of variability and imprecision in assessing the risk of groundwater contamination. Journal of contaminant hydrology, 93(1):72-84.

Dempster, A. P., 1967, "Upper and lower probabilities induced by a multivalued mapping," Annals of Mathematical Statistics, 38, pp. 325–339.

Rohmer, J., and Verdel, T.: Joint exploration of regional importance of possibilistic and probabilistic uncertainty in stability analysis. Computers and Geotechnics, 61, pp.308-315, 10.1016/j.compgeo.2014.05.015, 2014.

Shafer, G., 1976, A Mathematical Theory of Evidence. Princeton University Press

**Reviewer #2**

The study 'Deep uncertainties in shoreline change projections: an extra-probabilistic approach applied to sandy beaches' explores an elegant method to deal with a combination of a aleatoric (intrinsic) and epistemic (deep) uncertainties. The methods are well explained and seem easy to apply and very helpful to better understand different kind of uncertainties. And so, the manuscript convincingly illustates the methods' attractiveness.

We thank the reviewer for her/his insightful comments that, we believe, will contribute to improve the clarity of our manuscript. Please find below our responses (in blue) and how the manuscript has been revised (preceded by an arrow). Note that lines numbering refer to the trackchanged manuscript.

Given the attractiveness of the method and well-written manuscript (as discussed in the last section), I am a little surprised that some major sources of (deep) uncertainties are not included. Especially, the choice to only use one set of sea-level projections that do not include high-end scenarios seems a little odd. There are multiple recent sea-level projections that explicitly included high-end contributions of the W-AIS (e.g. Le Bars et al. 2016, Wong et al., 2017).

Thank you for your comment. We agree that we need to be more explicit when describing the design of the SLR projections used in our modelling framework. In fact, the possibility distributions of SLR projections do consider a high-end scenario that include large contributions of the W-AIS. The core of the trapezoidal possibility function for the SLR projections corresponds to the IPCC-SROCC likely range, to which we assign a possibility degree of 1. The boundaries of the support correspond to the low-end (lower limit) and high-end (upper limit) regional projections published by Le Cozannet et al. (2019) and Thiéblemont et al. (2019), respectively, to which we assign a possibility degree of 0. Hence, all SLR projections between the low-end and the high-end estimate have a non-zero possibility value, and are therefore considered as possible although very unlikely. It is important to note that both low-end and high-end scenarios have been designed so that they consider only physical-based modelling outcomes; i.e. we do not include expert judgement (e.g. Bamber et al., 2019). As shown in Table 2, the high-end estimates for the RCP8.5 scenario in 2100 for both sites reach SLR values larger than 1.8 m; this value appears to lie well within the projections of Wong et al. (2017) and Le Bars et al. (2017). Our high-end design is however very different from those of Wong et al. (2017) or Le Bars et al. (2017).

More details on the design of our high-end scenario are given in Thiéblemont et al. (2019). In short, we consider, for each contribution, the highest physically-based modelled global estimate that we could obtain from the literature and downscale it at regional scale using barystatic-fingerprints. For the sterodynamic contribution, MIROC5 and ACCESS1-0 are found to provide the largest contributions. For the glaciers component, the largest estimate (for the RCP8.5) was obtained by forcing a glacier model with HADGEM-ES (see Marzeion et al. 2012). For the Greenland component, we followed the largest model estimates of Fürst et al. (2015) that we corrected by the fact that CMIP5 models projection may underestimate future Greenland contribution since some atmospheric circulation patterns are not well represented (Delhasse et al., 2018). Finally, for the W-AIS contribution, we consider a mean projection assuming MICI, but not a worst-case model outcome because the confidence in MICI projection is still debated, and it is unsure that it will be initiated over the 21$^{st}$ century. In their former paper, DeConto and Pollard (2016) estimated that MICI could contribute to global sea-level

rise to more than 1 m by 2100. More recently, Edwards et al. (2019) revisited the latter results by considering the full range of uncertainties of the ice-sheet model parameters used by DeConto and Pollard (2016). The statistical treatment by Edwards et al. (2019) led to revise downward the DeConto and Pollard (2016) projection to 0.8 m. The latter value 0.8 m is hence used for our high-end scenario.

⇨ In the revised version of the manuscript, we expanded section 3.2 (L320-336) to make the design of our high-end scenarios clearer and slightly modified section 4.2 (L399-408)

In summary, we do consider several deep uncertainty sources in the design of our high-end scenario. In our study, we tried to be consistent and used a similar approach to define low-end and high-end scenarios (i.e. based on physically-based estimates). Nonetheless, we recognize that there exists several approaches (and studies) that have designed other sea-level high-end scenarios to which our possibility (and flexible) modelling framework could well be adapted.

Finally, to illustrate how flexible our framework is, we provide below an example of shoreline change projections where the possibility distribution for SLR projections is defined as a set of consecutive intervals instead of a trapezoidal distribution (see Figure R4a). These intervals correspond to the global mean sea-level projections of Le Cozannet et al. (2017) but considering the SROCC likely range instead of AR5. A possibility degree of 1 is assigned to the likely range (0.61-1.10 m). The review performed by Le Cozannet et al. (2017) reminded that 3 upper bounds for 2100 sea-level rise can be considered: 1.5 m, l 2 m or 5 m, to which a weight of 0.5, 0.4 and 0.1 is assigned, respectively, to reflect a lack of consensus in the scientific community regarding the maximum possible contribution of ice-sheets over the coming century. Le Cozannet et al. (2017) quote: "Referring to the IPCC terminology, we note that a 'medium degree of agreement' exists for maximum values of 1.5 m or 2 m, whereas a maximum value of 5 m is characterized by a 'low degree of agreement'", which translates into various degree of possibility shown in Figure R4a. Figure R4b shows the results applied to shoreline change projections. They reveal that including several "high-end" sea-level scenario leads to a strong enhancement of the gap between the lower and upper CDF near the upper tail of the probability box.

[Figure]

[Figure]

***Figure R4.*** *(b) Projected shoreline change probability box by 2100 for Aquitaine under the RCP8.5 scenario, by including (a) multiple global mean sea-level high-end projections prescribed through a possibility distribution consistent with the review of Le Cozannet et al. (2017).*

⇨ We now added a paragraph in the section 5.3 (L546-557) to reflect this.

**Minor:**

- the applied sea-level projections need a little more explanation. Do I understand well that 21 minus 2 CMIP projections were used? Could you explain in one sentence why two runs were judged 'unrealistic' with respect to sterodynamic behaviour (and others not). And could you explain how sources of uncertainty other than sterodynamical were included (or were they excluded)?

Concerning the sterodynamic component, we indeed discarded two models out of the 21 models of CMIP5; MIROC-ESM and MIROC-ESM-CHEM. Figure R5 below shows that MIROC-ESM (grey) and MIROC-ESM-CHEM (green) models project anomalously large sea-level rise in the Atlantic and North Sea areas. If these two models are discarded, the distribution obtained by the 19 remaining CMIP5 models in these areas is no longer significantly different from a Gaussian distribution according to the Shapiro–Wilk normality test. Furthermore, by 2100, the global-mean thermosteric sea-level rise of these two models (0.5 m for the RCP8.5 scenario) exceeds the median global-mean thermosteric sea-level rise of all other models (0.3 m) beyond 5 sigma (see Figure 3 of Le Cozannet et al., 2019). Finally, the CMIP5 historical MIROC-ESM and MIROC-ESM-CHEM simulations revealed unrealistic sea-surface height values of -15 m in the Mediterranean area that may suggest important biases in the regional sea-level calculations in these two models (Landerer et al., 2014).

[Figure]

***Figure R5.*** *CMIP5 sterodynamic projections in 2099 (ref period 1986–2005) for the North-Atlantic-N, North-Atlantic-S, Bay of Biscay, North Sea, Baltic Sea, Mediterranean-E, and Mediterranean-W Sea under the RCP8.5 scenario. Whisker boxes display the multi-model 1st*

*quartile, median, and 3rd quartile and the dashed line shows the multi-model mean. After Thiéblemont et al. (2019)*

Concerning the other contributions, i.e. glaciers, ice-sheet, landwater and GIA, uncertainties are indeed accounted for although we recognize that this was not made very clear in the current version of the manuscript near lines L283-298. The uncertainty of the mean sea-level is computed as the square root of the sum of the squares of each component uncertainty downscaled regionally. Note, however, that contributions that correlate with global-mean air temperature, namely the sterodynamic and ice-sheet surface mass balance components, have correlated uncertainties and are therefore added linearly (see Church et al., 2013, for more details). This procedure provides the regional mean sea-level IPCC likely-range to which we assign a possibility degree of 1 (see L359-370 and table 2). Nonetheless, the likely-range does not cover the full uncertainty range; that is why we considered low-end and high-end estimates to bound the support of the trapezoid (see also response to first comment).

⇨ We revised section 3.2 substantially (L293-L319). It now better describes how our projections were constructed and what the main assumptions are.

- In figure 4, could you explain how the linear fit was made? In 4b, the fist black dot (yr=1989, shore line change ~ -491) seems to be excluded. Otherwise, I would expect a very different R2.

Thank you for noticing this point, which indeed deserves some clarification. Here we focus on annual values, so we apply the linear regression on annual means. However, given the irregularity of the temporal sampling, the number of points per year can vary widely; especially, recent periods are more covered than past periods. To account for this irregularity in the sampling, the regression is weighted by the number of samples per year. In that respect, the year 1989, which is represented by only 3 samples, has a very small weight. Removing the weighting procedure would lead to reduce drastically the $R^2$ (it falls below 0.40) but impacts the trend coefficients only modestly (we obtain 0.40(0.10) m/y instead of 0.60(0.07) m/y).

⇨ This has been clarified (L176-179)

**References:**

Bamber, J. L., Oppenheimer, M., Kopp, R. E., Aspinall, W. P., and Cooke, R. M.: Ice sheet contributions to future sea-level rise from structured expert judgment, Proceedings of the National Academy of Sciences, 116, 11195, 10.1073/pnas.1817205116, 2019

Church, J. A., Clark, P. U., Cazenave, A., Gregory, J. M., Jevrejeva, S., Levermann, A., Merrifield, M. A., Milne, G. A., Nerem, R. S., Nunn, P. D., Payne, A. J., Pfeffer, W. T., Stammer, D., and Unnikrishnan, A. S.: Sea Level Change, In: Climate Change 2013: The Physical Science Basis. Contribution of Working Group I to the Fifth Assessment Report of the Intergovernmental Panel on Climate Change ed., Cambridge University Press, Cambridge, United Kingdom and New York, NY, USA., 2013.

Delhasse, A.; Fettweis, X.; Kittel, C.; Amory, C.; Agosta, C. Brief communication: Impact of the recent atmospheric circulation change in summer on the future surface mass balance of the Greenland Ice Sheet. Cryosphere 2018, 12, 3409–3418.

DeConto, R.M.; Pollard, D. Contribution of Antarctica to past and future sea-level rise. Nature 2016, 531, 591–597.

Edwards, T.L.; Brandon, M.A.; Durand, G.; Edwards, N.R.; Golledge, N.R.; Holden, P.B.; Nias, I.J.; Payne, A.J.; Ritz, C.; Wernecke, A. Revisiting Antarctic ice loss due to marine ice-cliff instability. Nature 2019, 566, 58.

Furst, J.J.; Goelzer, H.; Huybrechts, P. Ice-dynamic projections of the Greenland ice sheet in response to atmospheric and oceanic warming. Cryosphere 2015, 9, 1039–1062

Landerer, F.W.; Gleckler, P.J.; Lee, T. Evaluation of CMIP5 dynamic sea surface height multi-model simulations against satellite observations. Clim. Dyn. 2014, 43, 1271–1283.

Le Bars, D., *et al* 2017 *Environ. Res. Lett.* **12** 044013

Le Cozannet, G., Manceau, J. C., and Rohmer, J.: Bounding probabilistic sea-level projections within the framework of the possibility theory, Environmental Research Letters, 12, 10.1088/1748-9326/aa5528, 2017.

Le Cozannet, G., Thiéblemont, R., Rohmer, J., Idier, D., Manceau, J.-C., and Quique, R.: Low-End Probabilistic Sea-Level Projections, Water, 11, 10.3390/w11071507, 2019.

Marzeion, B.; Jarosch, A.H.; Hofer, M. Past and future sea-level change from the surface mass balance of glaciers. Cryosphere 2012, 6, 1295–1322.

Thiéblemont, R., Le Cozannet, G., Toimil, A., Meyssignac, B., and Losada, I. J.: Likely and High-End Impacts of Regional Sea-Level Rise on the Shoreline Change of European Sandy Coasts Under a High Greenhouse Gas Emissions Scenario, Water, 11, 10.3390/w11122607, 2019

Wong, T.E., Bakker, A.M.R. & Keller, K. Impacts of Antarctic fast dynamics on sea-level projections and coastal flood defense. Climatic Change 144, 347–364 (2017). https://doi.org/10.1007/s10584-017-2039-4